# Molecular characterization of renal cell carcinoma tumors from a phase III anti-angiogenic adjuvant therapy trial

Robert J. Motzer [1,9] ✉, Jean-François Martini[2,9], Xinmeng J. Mu[3], Michael Staehler [4], Daniel J. George [5], Olga Valota [6], Xun Lin [2], Hardev S. Pandha[7], Keith A. Ching[3,10] & Alain Ravaud[8,10]

Multigene assays can provide insight into key biological processes and prognostic information to guide development and selection of adjuvant cancer therapy. We report a comprehensive genomic and transcriptomic analysis of tumor samples from 171 patients at high risk for recurrent renal cell carcinoma post nephrectomy from the S-TRAC trial (NCT00375674). We identify gene expression signatures, including STRAC11 (derived from the sunitinib-treated population). The overlap in key elements captured in these gene expression signatures, which include genes representative of the tumor stroma microenvironment, regulatory T cell, and myeloid cells, suggests they are likely to be both prognostic and predictive of the anti-angiogenic effect in the adjuvant setting. These signatures also point to the identification of potential therapeutic targets for development in adjuvant renal cell carcinoma, such as MERTK and TDO2. Finally, our findings suggest that while anti-angiogenic adjuvant therapy might be important, it may not be sufficient to prevent recurrence and that other factors such as immune response and tumor environment may be of greater importance.

Understanding the key biologic processes involved in micrometastasis dissemination, colonization, nesting, dormancy, immune evasion/suppression, and tumor microenvironment modulation remains critical in identifying the right drug mode of action for adjuvant therapy in patients at high risk of disease recurrence[1,2]. Patients with localized (stage I–III) renal cell carcinoma (RCC) commonly undergo radical or partial nephrectomy with curative intent, but ~60% of the patients at high risk of disease recurrence, based on a validated nomogram, will relapse[3]. Effective adjuvant treatment to prevent recurrence of RCC is an unmet need.

Clear cell RCC (ccRCC) tumors, the most common histological RCC subtype, are highly vascularized. These tumors frequently (>70%) present with biallelic inactivation of the von Hippel-Lindau (VHL) gene[4]. VHL inactivation leads to the stabilization and accumulation of hypoxia-inducible factors (HIF-1α and HIF-2α). These transcription factors activate numerous downstream targets, including vascular endothelial growth factor (VEGF), the foremost growth factor in tumor angiogenesis. The VEGF pathway contributes to the development and metastasis of cancer in several additional ways. VEGF receptor-1 (VEGFR1)-positive hematopoietic progenitor cells (HPCs) and bone marrow-derived cells play important roles in the colonization of

[1]Department of Medicine, Memorial Sloan Kettering Cancer Center, New York, NY 10065, USA. [2]Global Product Development-Oncology, Pfizer Inc, La Jolla, CA 92121, USA. [3]Oncology Research Unit, Pfizer Worldwide Research and Development Medicine, Pfizer Inc, La Jolla, CA 92121, USA. [4]Department of Urology, University Hospital of Munich, Munich, Bavaria 80333, Germany. [5]Department of Medicine, Duke Cancer Institute, Durham, NC 27710, USA. [6]Global Product Development-Oncology, Pfizer S.r.L, Milan, Lombardy 20152, Italy. [7]Department of Medical Oncology, University of Surrey, Guildford, England GU2 7XS, UK. [8]Department of Medical Oncology, Bordeaux University Hospital, Bordeaux 33300, France. [9]These authors contributed equally: Robert J. Motzer, Jean-François Martini. [10]These authors jointly supervised this work: Keith A. Ching, Alain Ravaud. ✉e-mail: motzerr@mskcc.org

premetastatic sites prior to tumor cell arrival[1]. The tumor cells that survive the initial immune defense can survive for years as dormant micrometastases. VEGFR1-positive HPCs and VEGFR2-positive endothelial progenitor cells are thought to mediate the neovascularization of these tumors[1]. VEGF contributes to immune evasion and suppression and can lead to changes in the tumor microenvironment via effects on T cells, regulatory T cells (Tregs), myeloid-derived suppressor cells, and dendritic cells[5,6].

All but one of the clinical trials with tyrosine kinase inhibitors (TKIs) of the VEGF pathway in the adjuvant RCC setting failed to show clinical benefit[7,8]. There are various hypotheses to explain why S-TRAC had a positive outcome while the other trials did not, including the overall drug exposure/dose density and the clinicopathologic selection of the patient population enrolled[7,9,10]. S-TRAC enrolled patients with ccRCC at high risk for recurrence post nephrectomy and demonstrated significantly longer disease-free survival (DFS) with adjuvant sunitinib compared with placebo[11]. Based on the S-TRAC results, sunitinib received regulatory approval in the United States as adjuvant treatment for adult patients at high risk of recurrent RCC following nephrectomy.

Using clinical data from the S-TRAC trial and biospecimens collected in the study, various approaches were deployed to identify and possibly enrich those patients more likely to benefit from such an adjuvant approach[12–15], including a 16-gene Recurrence Score® assay that looked at components of vasculature normalization, immune response, inflammation, and cell growth and differentiation, and that displayed strong prognostic performance but did not predict differential benefit from adjuvant therapy[15].

Multigene assays have been shown to provide prognostic, and sometimes predictive, information beyond traditional clinical parameters relevant for selection of adjuvant therapy in several tumor types. Hence, we conducted comprehensive, integrated multi-omics tumor analyses in a subset of 171 patients ($n = 91$ and 80 treated with sunitinib and placebo, respectively) from the S-TRAC trial to identify molecular subtypes associated to these patients, and potential targets or pathways of interest that may inform treatment or combination therapies for patients with RCC who are at high risk of recurrence following nephrectomy. Furthermore, we looked for possible mechanisms of resistance to TKIs in this setting.

In this work, we utilize tumor tissue obtained prospectively from a large, global phase III trial of patients rendered disease-free by nephrectomy, who are at high risk of RCC recurrence and treated with adjuvant anti-angiogenic therapy (sunitinib) or placebo, we identify three gene expression signatures (GES) consisting of genes representative of the tumor stroma microenvironment that, when highly expressed, are associated with poor outcome and short DFS. We show that the overlap of elements in these three GES suggests that they are likely to be prognostic and predictive of the anti-angiogenic effect in the adjuvant setting, and we identify potential therapeutic targets for development in adjuvant RCC.

## Results

### Classic ccRCC mutations do not predict outcome following tumor resection

As part of a retrospective hypothesis-generating analysis, we submitted 171 tumor tissues from nephrectomy or biopsy of patients enrolled in the S-TRAC trial (NCT00375674) to whole exome sequencing (WES) to identify and, when possible, compare genetic correlates of clinical outcome with the results reported elsewhere in the metastatic setting. Molecular characterization of ccRCC has led to the identification of commonly mutated genes, including *PBRM1*, *SETD2*, and *BAP1*. After *VHL*, *PBRM1* is the most commonly mutated gene in ccRCC, which, depending on context, can act as a tumor suppressor or oncogene[16]. A meta-analysis of seven studies in localized RCC found that *PBRM1* loss of function was associated with reduced overall

survival (OS) and progression-free survival (PFS), as well as advanced clinicopathologic features[17].

Applying WES to the samples from our cohort confirmed the prevalence of alterations in *VHL*, *PBRM1*, *SETD2*, and *BAP1* (Fig. 1a). The prevalence of these alterations was comparable to those reported in the IMmotion 150, IMmotion 151, and JAVELIN Renal 101 studies[18–20]. However, DFS was not significantly influenced by the presence of mutations in *VHL*, *PBRM1*, *SETD2*, and *BAP1* in the overall study population (Fig. 1b–d). Loss of expression of *VHL* is a hallmark of ccRCC and, as expected, mutations in *VHL* were detected in the majority of patients overall (64.9%) (Fig. 1a). Mutations in *PBRM1* were the second-most prevalent (30.4%) (Fig. 1a). Although patients with tumors harboring a *PBRM1* mutation seem to have a shorter DFS versus patients with wild-type tumor (Fig. 1b), *PBRM1* mutations did not appear to differentiate DFS outcomes in treatment-specific arms (Fig. 1e). As suggested by previous molecular analysis of early stage ccRCC, although mutations in *PBRM1* are frequent, they may not differentiate aggressive versus non-aggressive tumors as much as represent an early, potentially essential event in tumorigenesis that does not impact significantly clinical outcome[21]. Conflicting results were reported in previously untreated patients with advanced metastatic (Stage IV) RCC[18–20,22,23]. In addition, expression signatures for cancer cell subpopulations and immune evasion are associated with *PBRM1* mutation and survival in primary and advanced RCC treated with checkpoint inhibitors[24,25]. Mutations in *BAP1*, another key tumor suppressor gene located on chromosome 3p near *SETD2*, *PBRM1*, and *SMARCC1* associated with poor prognosis in many cancers[26], were rare (12.9%) (Fig. 1a). These mutations were not associated with a numerically shorter DFS in the adjuvant setting (Fig. 1c, e), which is similar to previous findings in the first-line metastatic setting[20], but small numbers could have limited this interpretation.

We also demonstrated that *MTOR* mutations were associated with poor prognosis (Fig. 1e, f). Activating somatic mutations of *MTOR* are known to occur at low frequency (~6%) and lead to hyperactive mTORC1 [mammalian target of rapamycin complex 1] signaling[27,28]; however, in our cohort of patients enriched with clinical characteristics of high risk of recurrence, the frequency of *MTOR* mutation was slightly higher, at 10.5% (Fig. 1a). Similarly, we observed treatment arm–specific differences in DFS relative to wild-type when mutations in *ARID1A* (with mutation frequency of 11.1%) were present (Fig. 1e, g). *ARID1A* encodes a protein that forms part of SWI/SNF [switch/sucrose non-fermentable] chromatin remodeling complex; however, a genome-scale RNAi- and CRISPR-Cas9 analysis classified *PBRM1* and *ARID1A* as separate functional modules, suggesting loss or mutation of these genes might not be mechanistically related[29].

### Mutations in immune-related or chromatin homeostasis genes influenced treatment outcome

We re-examined the WES data to analyze functional relationships among the genes associated with differential DFS between mutated and non-mutated tumors for the specific gene of interest in the S-TRAC study. A set of variants in either *THEMIS*, *WDFY4*, or *CSPG4* that did not predict outcome in the overall population but were associated with longer DFS in sunitinib-treated patients were also identified (Fig. 1h, i). Interestingly, these three genes are connected to T-cell activity and maintenance: *THEMIS* has been shown to be required for peripheral CD8+ T-cell maintenance[30,31]; *WDFY4* for antitumor immunity by activating immunological T cells[32,33]; and *CSPG4* for activation, maturation, proliferation, and migration of different immune cell subsets[34]. In addition, mutations in *CTCFL*, *KMT2D*, *PPIP5K1*, and *ERICH6B* were also associated with longer DFS in sunitinib-treated patients (Fig. 1I); of note, the main variant allele detected and driving the effect for *ERICH6B*, pS174T, is actually a germline variant. *CTCFL* may be required to promote resistance

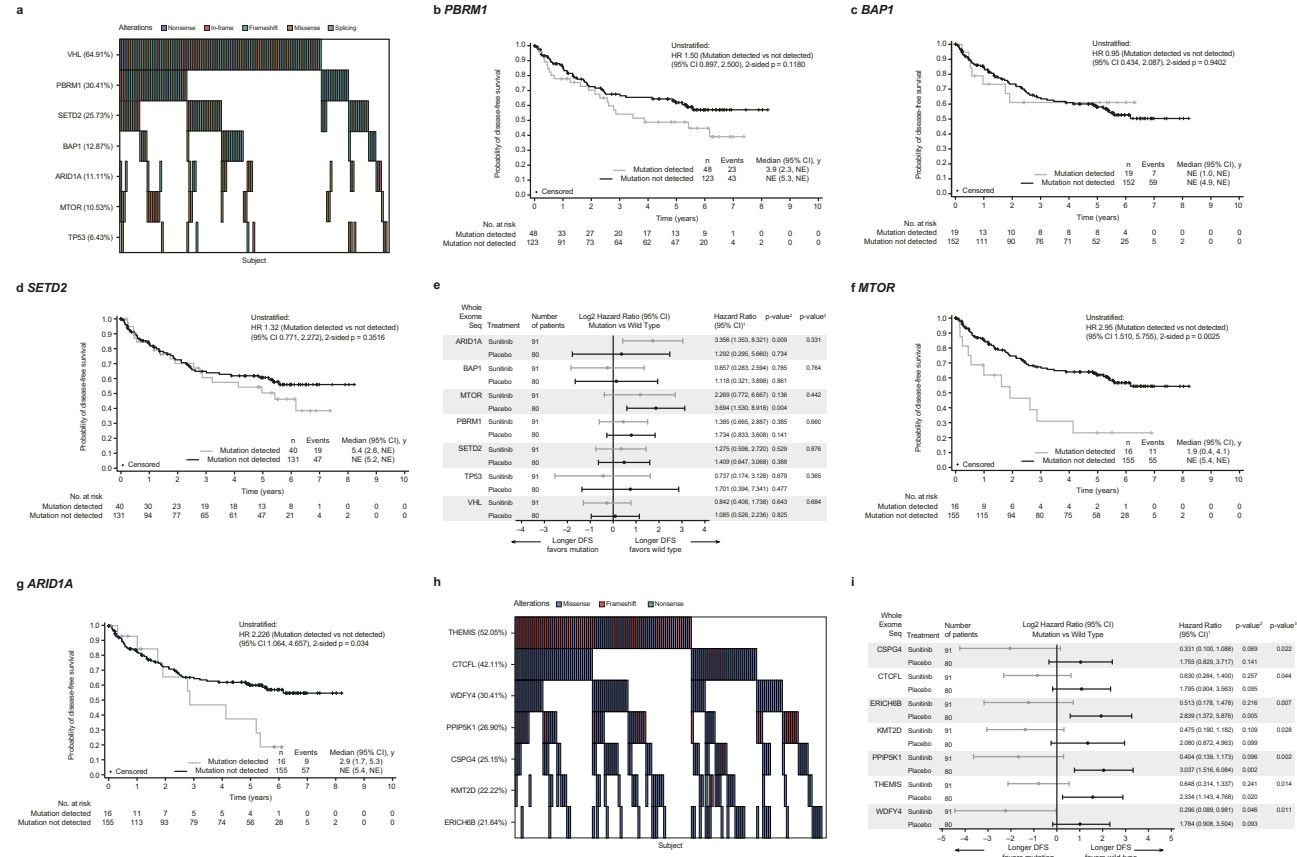

**Fig. 1 | Mutational analysis of selected genes. a** Heatmap of selected genes based on whole exome sequencing. Kaplan–Meier plots stratified by mutation vs mutation not detected in the overall cohort for **b** *PBRM1*; **c** *BAP*; and **d** *SETD2*. **e** Forest plot for treatment arm effect of gene mutations and DFS association stratified by mutation vs mutation not detected. Data are presented as hazard ratio (HR) and 95% CI. Kaplan–Meier plots stratified by mutation vs mutation not detected in the overall cohort for **f** *MTOR*; and **g** *ARID1A*. **h** Heatmap of gene mutations that influenced sunitinib treatment outcome based on whole exome sequencing. **i** Forest plot for treatment arm effect of additional gene mutations and DFS association stratified by mutation vs mutation not detected. Data are presented as HR and 95% CI. HR <1 indicates longer DFS in mutation group; HR >1 indicates longer DFS in mutation not detected group. [1]Cox proportional hazards model with <median as the reference group was used to calculate HR and 95% CI. [2]Cox regression HR *p*-value is used to compare between Wild Type/Mutation groups. A

HR <1 indicates better survival in the Mutation group, while a HR >1 indicates better survival in the Wild Type group. HR reference level is <median, *p*-value is from Logrank test. [3]Two-sided *p*-value for overall Wild Type/Mutation-by-treatment interaction from Cox model with treatment group and wild-type/mutation status as two independent variables. ARID1A AT-rich interaction domain 1A, BAP1 BRCA1-associated protein 1, CI confidence interval, CSPG4 chondroitin sulfate proteoglycan 4, CTCFL CCCTC-binding factor-like, DFS disease-free survival, ERICH6B glutamate-rich 6B, HR hazard ratio, KMT2 lysine methyltransferase 2D, MTOR mechanistic target of rapamycin kinase, NE not estimable, PBR1 proline-rich protein BstNI subfamily 1, PPIP5K1 diphosphoinositol pentakisphosphate kinase, SETD2 SET domain-containing 2, histone lysine methyltransferase, THEMIS thymocyte selection-associated, TMB tumor mutational burden, TP53 tumor protein 53, VHL Von Hippel-Lindau tumor suppressor, WDFY4 WDFY family member 4.

phenotype through formation of BORIS-regulated alterations in chromatin[35,36], *KMT2D* for the making of histone methyltransferase[37], and *PPIP5K1* for regulating cell motility[38].

## High tumor mutational burden is associated with poor prognosis

As previously established in the context of metastatic disease, low tumor mutational burden (TMB) versus high TMB was prognostic of better survival outcomes[39]. In this adjuvant setting, low versus high TMB (based on median cutoff, without using a pre-set cutoff of 10 or 20 mutations per Mb) was associated with a numerically longer DFS in the overall population (Fig. 2a). This effect was mainly driven by the placebo arm (hazard ratio [HR] 2.89; 95% confidence interval [CI]: 1.40–5.93; *p* = 0.0052; Fig. 2b). Interestingly, treatment with sunitinib in the adjuvant setting largely abrogated this prognostic effect (HR 0.76 [95% CI: 0.37–1.55]; *p* = 0.5619; Fig. 2b), suggesting that high TMB may be confounded with a clinical benefit to adjuvant sunitinib.

## Previously defined immune and angiogenic-associated transcriptomic signatures predict outcome in the overall cohort population

Next, we investigated previously defined GES established in the advanced metastatic setting (Stage IV) in the two studies, IMmotion 150 and JAVELIN Renal 101, in the subset of 133 cases that were successfully profiled in the S-TRAC trial. These two studies represent a different patient population compared with the patients in the S-TRAC trial who had locally advanced, completely resected primary tumors (Stage III); however, both IMmotion 150 and JAVELIN Renal 101 utilized sunitinib (standard of care at the time for advanced/metastatic stage disease) as comparator arm. The IMmotion 150 trial compared atezolizumab (programmed cell death ligand 1 [PD-L1] inhibitor) and atezolizumab plus bevacizumab (anti-angiogenesis agent) versus sunitinib, and the JAVELIN Renal 101 trial compared axitinib (anti-angiogenesis agent) plus avelumab (PD-L1 inhibitor) versus sunitinib[18,20]. In the overall S-TRAC study population, low versus high expression of myeloid inflammation GES (IMmotion 150

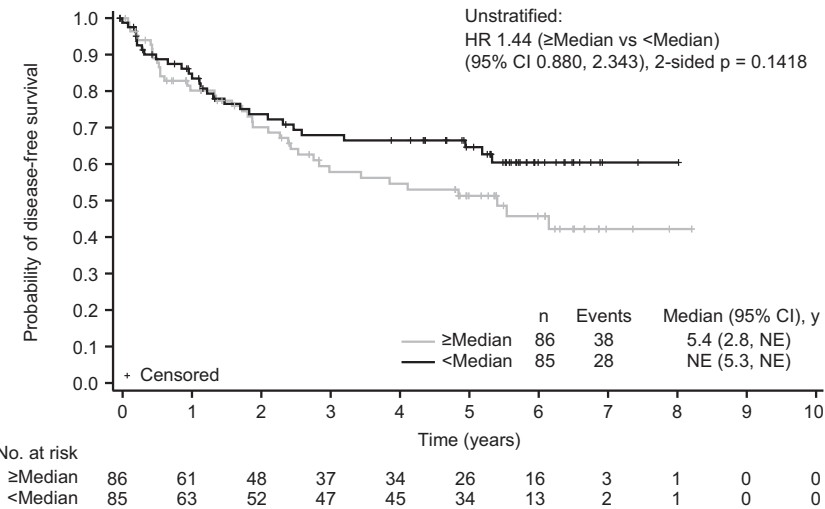

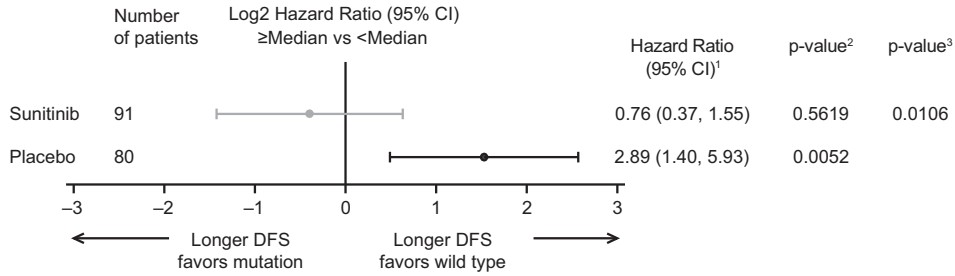

**Fig. 2 | DFS association with TMB. a** Kaplan–Meier plot for TMB in the overall cohort and **b** treatment arm effect of TMB and DFS association stratified by median cutoff. Data are presented as HR and 95% CI. [1]Cox proportional hazards model with <Median as the reference group was used to calculate HR and 95% CI. [2]Log-rank two-sided test was performed to compare between overall median cutoff groups. A HR <1 indicates better survival in the ≥Median group, while a HR >1 indicates better survival in the <Median group. HR reference level is <median, p-value is from Logrank test. [3]Two-sided p-value for overall median cutoff-by-treatment interaction from Cox model with treatment group and median cutoff status as two independent variables. CI confidence interval, DFS disease-free survival, HR hazard ratio, NE not estimable, TMB tumor mutational burden.

myeloid inflammation GES) was associated with longer DFS (HR 2.22 [95% CI: 1.27–3.88]) (Fig. 3a); the effect was further accentuated in the sunitinib-treated group, whereby longer DFS was associated with lower expression of the myeloid inflammation GES (HR 3.60 [95% CI: 1.62–8.03]) (Fig. 3b). High versus low expression of the T-cell effector (Teff) GES (IMmotion 150 Teff GES) was associated with a numerically longer DFS in the overall S-TRAC study population (HR 0.69 [95% CI: 0.40–1.19]) (Fig. 3c). Similarly, high versus low expression of the Immune GES established in the JAVELIN Renal 101 study (JAVELIN Renal 101 Immune GES) was associated with longer DFS in the overall S-TRAC study population (HR 0.53 [95% CI: 0.30–0.92]) (Fig. 3d). The JAVELIN Renal 101 Immune GES consists of 26 genes that comprise regulators of both adaptive and innate immune responses (T cell and natural killer [NK] cells), cell trafficking, and inflammation but displayed limited overlap with the IMmotion 150 Teff signature[20].

With application of the IMmotion 150 angiogenesis GES to the S-TRAC samples, a high versus low expression of angiogenesis GES was associated with a numerically longer DFS with HR: overall 0.64 (95% CI 0.37–1.10); sunitinib 0.71 (95% CI 0.34–1.50); and placebo 0.56 (95% CI: 0.25–1.28) (Fig. 3b, e). This association was further observed and somewhat stronger when we used the similar yet distinct (with only *CD34* and *KDR* present in both signatures) angiogenesis GES that was derived from the JAVELIN Renal 101 study[20]: overall HR 0.53 (95% CI:

0.31–0.93); sunitinib HR 0.52 (95% CI: 0.25–1.08); placebo HR 0.46 (95% CI 0.20–1.07) (Fig. 3b, f). In the JAVELIN Renal 101 trial, angiogenesis GES was significantly associated with longer PFS in the sunitinib arm and it did not differentiate PFS in the combination arm (avelumab plus axitinib) in the setting of metastatic RCC.

## The Cancer Genome Atlas Kidney Renal Clear Cell Carcinoma classification, immune cell-type specific gene expression profiles, and other metabolic signatures

Based on unsupervised clustering methods, four stable messenger RNA (mRNA) expression subtypes (m1–m4) have been previously identified by The Cancer Genome Atlas Kidney Renal Clear Cell Carcinoma (TCGA KIRC) in their RCC cohort[27]. Applying this classification to the S-TRAC overall population showed a stratified DFS with an order of m1, m2, m4, and m3 from longest to shortest DFS (Fig. 4a. Similarly, classifying the patients from the JAVELIN Renal 101 study had an order of m1, m2, m4, and m3 from longest to shortest PFS in the sunitinib treatment arm[20]. The m3 group, which is enriched with deletion of *CDKN2A* and mutations in *PTEN* and is associated with a short OS[27], has a worse prognosis with sunitinib in all three studies (Fig. 4b)[20,27].

To further characterize the tumor microenvironment composition of the samples from the patients enrolled in S-TRAC, we inferred 64 cell types using the xCell transcriptional signature-based method

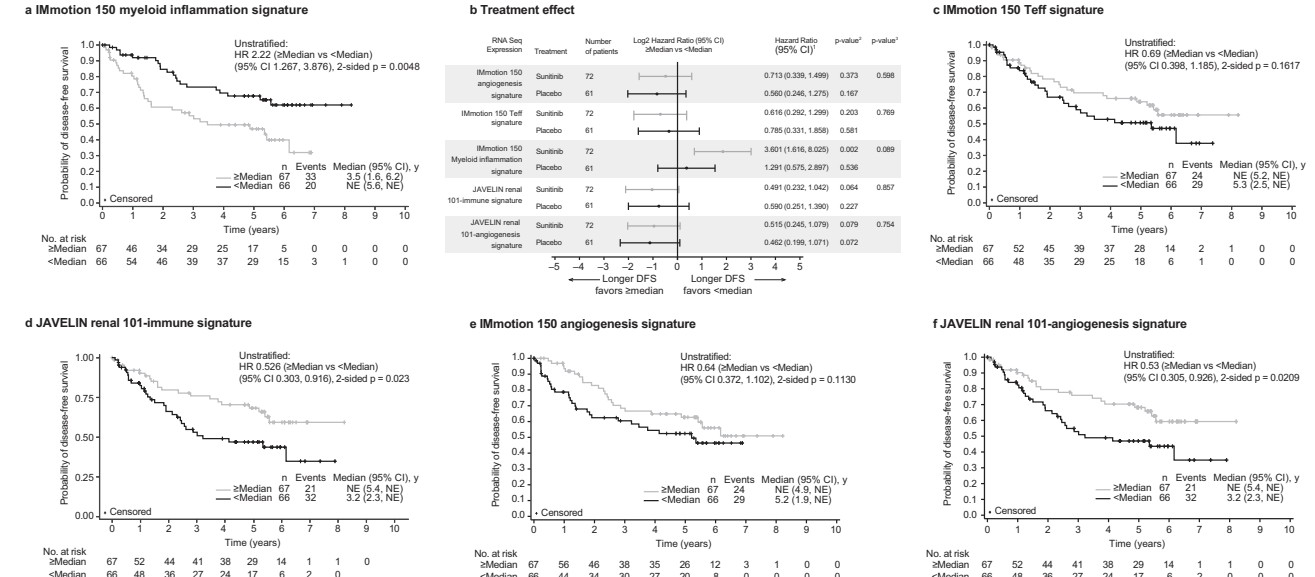

**Fig. 3 | DFS according to previously defined gene expression signatures.**
**a** Kaplan–Meier plot in the overall population for the IMmotion 150 myeloid inflammation signature; **b** Forest plot with treatment arm effect for the association of RNA sequence expression and DFS stratified by median cutoff; data are presented as HR and 95% CI; Kaplan–Meier plot in the overall population of **c** IMmotion 150 Teff signature; **d** JAVELIN Renal 101-Immune signature; **e** IMmotion 150-Angio Signature; and **f** JAVELIN Renal 101-Angio signature. HR <1 indicates longer DFS in the ≥Median group; HR >1 indicates longer DFS in the <Median group. [1]Cox proportional hazards model with <median as the reference group was used to

calculate HR and 95% CI. [2]Cox regression HR *p*-value is used to compare between overall median cutoff groups. A HR <1 indicates better survival in the ≥Median group, while a HR >1 indicates better survival in the <Median group. HR reference level is <median, *p*-value is from Logrank test. [3]Two-sided *p*-value for overall median cutoff-by-treatment interaction from Cox model with treatment group and median cutoff status as two independent variables. HR analyses are adjusted for sex and age (<65, ≥65), using proportional hazards modeling. No adjustments were made for multiple comparisons. Angio angiogenesis, CI confidence interval, DFS disease-free survival, HR hazard ratio, NE not estimable, Teff effector T cell.

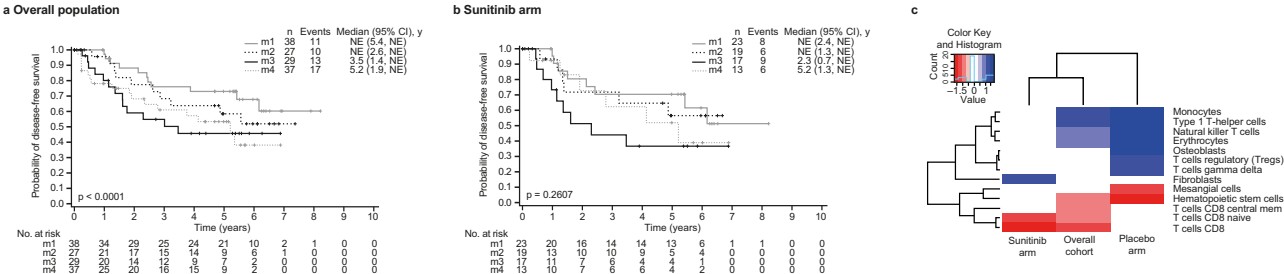

**Fig. 4 | S-TRAC trial samples classification according to the TCGA KIRC molecular subtypes and S-TRAC trial samples immune cell-type–specific gene expression profiles.** Kaplan–Meier plots of DFS stratified according to TCGA molecular subtype classification for **a** the overall population and **b** sunitinib treatment arm. **c** Cell types significantly associated with DFS in S-TRAC trial samples. Longer DFS (red), shorter DFS (blue), and non-significant (white). Each cell type score (xCell) was stratified by >median or ≤median for the overall

cohort, sunitinib arm, or placebo arm. The likelihood ratio was filtered for *p* < 0.05 and the log2(HR) was clustered in a heatmap. Non-significant cell types for both arms and overall cohort are not shown. For significant and non-significant cell types for both arms and overall cohort, xCell scores are provided in the file Figure4c_celltypes_DFS/xCell.complete.tsv CI confidence interval, DFS disease-free survival, HR hazard ratio, NE not estimable, TCGA The Cancer Genome Atlas.

(Fig. S1)[40]. We examined these signatures in the context of DFS in the overall S-TRAC study and then by treatment arm (Fig. 4c). In the overall cohort, monocytes, type 1 T-helper, and NK T cells were associated with worse DFS, whereas CD4+ T cells, CD8+ T cells, CD8+ central memory T cells, multipotent progenitors, erythrocytes, and hematopoietic stem cells were associated with prolonged DFS (*p* < 0.05). Interestingly, when focusing on the sunitinib arm alone, fibroblasts were associated with worse DFS, whereas NK, CD4+, and CD8+ T cells were associated with better DFS. In the placebo arm, NK T cells, monocytes, osteoblasts, Tregs, and type 1 T-helper signatures were associated with shorter DFS. In contrast, hematopoietic stem and mesangial cell signatures were associated with longer DFS. In addition, although monocytes were strongly associated with shorter DFS except

in the sunitinib arm, in contrast, fibroblasts were associated with shorter DFS in the sunitinib arm only.

## Elastic net combinatorial biomarker approach identified transcriptomic signatures associated with high risk of recurrence and poor prognosis

The application of the GES established in advanced and metastatic RCC to the DFS data from S-TRAC trial in patients with RCC at high risk of recurrence post nephrectomy showed the importance of the angiogenic component as well as the immune component of the tumors and microenvironments. We therefore explored the development of GES that would be relevant to lower-stage disease, with the intent to further identify patients who are at the highest risk for relapse

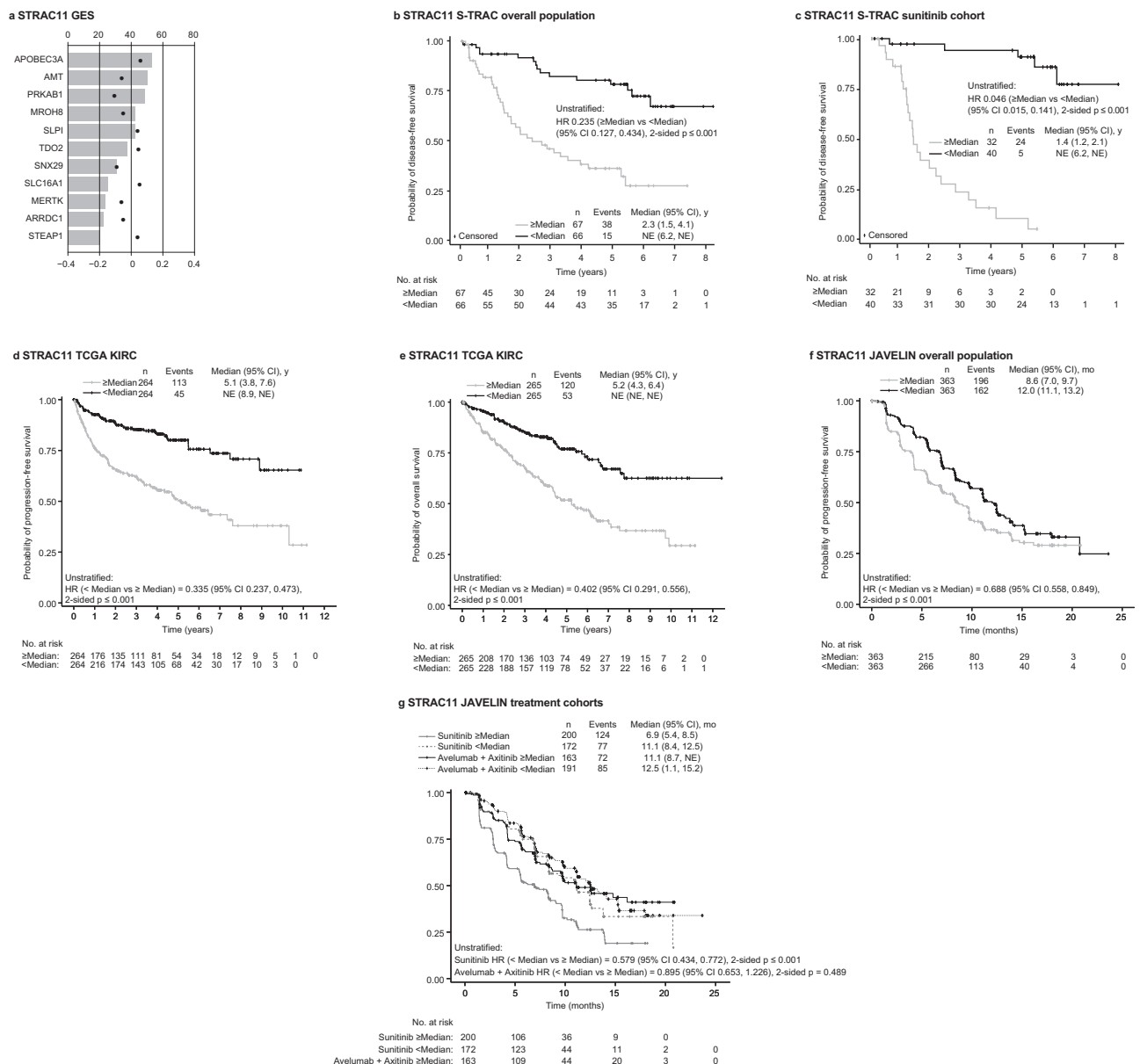

**Fig. 5 | Identification of transcriptomic signatures associated with a high risk of recurrence and poor prognosis. a** Discovery of the STRAC11 GES [derived from the sunitinib-treated S-TRAC trial population] according to bootstrapping frequency. Kaplan–Meier plot of **b** differential DFS probability by STRAC11 in the S-TRAC placebo arm and **c** sunitinib arm. Kaplan–Meier plot of differential **d** PFS probability and **e** OS probability by STRAC11 in the TCGA data set; **f** differential PFS by STRAC11 in the JAVELIN Renal 101 overall population data set and **g** by treatment arm. AMT aminomethyltransferase, APOBEC3A apolipoprotein B mRNA editing enzyme catalytic subunit 3A, ARRDC1 arrestin domain-containing 1, CI confidence interval, DFS disease-free survival, HR hazard ratio, MERTK MER tyrosine kinase, MROH8 maestro heat like repeat family member 8, NE not estimable, PRKAB1 protein kinase AMP-activated non-catalytic subunit beta 1, SLC16A1 solute carrier family 16, member 1, SLPI secretory leukocyte peptidase inhibitor, SNX29 sorting nexin 29, STEAP1 six transmembrane epithelial antigen of the prostate family member 1, TCGA The Cancer Genome Atlas, TDO2 tryptophan 2,3-dioxygenase.

based on their clinicopathologic characteristics and could potentially benefit from additional combination therapy strategies for this setting.

First, we focused on the sunitinib treatment arm of the S-TRAC study to identify molecular characteristic that might associate with the resistance to anti-angiogenesis adjuvant treatment and could be predictive in nature. Using the sunitinib treatment arm dataset only, we identified a GES signature that consists of 11 genes (STRAC11): *APO-BEC3A, AMT, PRKAB1, MROH8, SLPI, TDO2, SNX29, SLC16A1, MERTK, ARRDC1*, and *STEAP1* (Fig. 5a). In the placebo arm, low and high STRAC11 GES expression groups had similar DFS (HR 0.8 [95% CI: 0.36–1.82]; Fig. 5b). In contrast, in sunitinib-treated patients, low versus high expression of the STRAC11 GES was associated with longer DFS (HR 0.05 [95% CI: 0.02–0.14]) (Fig. 5c). Following the co-expression and network analyses, this signature was found to be enriched for genes involved in the regulation of the stroma component of the tumor (*TDO2, STEAP1*) as well as Treg cells (*SLC16A1, PRKAB1*) and myeloid cell (*APOBEC3A, MERTK, SNX29*) subsets (Table 1).

To verify its strength, the STRAC11 GES signature was applied to two independent datasets in the advanced RCC setting: the TCGA KIRC dataset (Cancer Genome Atlas Research, 2013) (Fig. 5d, e) and JAVELIN Renal 101 metastatic RCC population (Fig. 5f, g). In the TCGA KIRC dataset, expression was particularly high in m3, and to a lesser extent, in m4 TCGA KIRC subtypes. As anticipated, low versus high expression of STRAC11 GES was associated with longer PFS and longer OS (Fig. 5d,

**Table 1 | Genes included in STRAC11, STRAC13, and STRAC14**

| | Gene | Class/Function |
|---|---|---|
| STRAC13 and STRAC11 | *AMT* | Aminomethyltransferase (Glycine Cleavage System Protein T) |
| STRAC13 | *CXCR1* | High Affinity Interleukin-8 Receptor A; recruits immune suppressive cells such as the myeloid-derived suppressor cells to the tumor microenvironment |
| | *ATG14* | Autophagy-Related Protein 14-Like Protein; regulates cytoprotective autophagy (role in drug resistance) |
| | *FPR2* | Formyl Peptide Receptor 2; promotes invasion and metastasis of some cancers |
| | *SPIN3* | Spindlin-Like Protein 3; a tumor suppressor, pro-apoptotic, downregulates *CYCD*1 |
| | *ZNF415* | Zinc Finger Protein 415; involved in transcriptional regulation; promoter aberrantly hypermethylated in oropharyngeal squamous cell carcinoma |
| | *THEM4* | Thioesterase Superfamily Member 4; facilitates apoptosis by negatively regulating Akt/PKB signaling |
| | *TXNIP* | Thioredoxin Interacting Protein; a tumor suppressor in various cancers; regulate the metabolism and division of cells by inhibiting their ability to take up glucose |
| | *ADHFE1* | Hydroxyacid-Oxoacid Transhydrogenase; a breast cancer oncogene |
| | *UFSP1* | UFM1-Specific Peptidase 1 (non-functional) |
| | *ZKSCAN7* | Zinc Finger With KRAB And SCAN Domains 7; may be involved in transcriptional regulation |
| | *BIRC7* | Baculoviral IAP Repeat Containing 7; inhibits apoptosis by inhibiting proteolytic activation of capsases |
| | *HIST2H3A* | H3 Clustered Histone 15; core component of nucleosome |
| STRAC11 | *APOBEC3A* | Apolipoprotein B MRNA Editing Enzyme Catalytic Subunit 3A; induces mutagenesis in cancer cells, and contributes to tumor evolution |
| | *PRKAB1* | Protein Kinase AMP-Activated Non-Catalytic Subunit Beta 1; a regulatory subunit of the AMP-activated protein kinase (AMPK), can act as either a tumor suppressor (prevent tumorigenesis) or a tumor promoter (after tumorigenesis occurred) |
| | *MROH8* | Maestro Heat Like Repeat Family Member 8 |
| | *SLPI* | Secretory Leukocyte Peptidase Inhibitor; modulates the inflammatory and immune responses and the promotion of cell proliferation |
| | *TDO2* | Tryptophan 2,3-Dioxygenase; may play a role in cancer through the suppression of antitumor immune responses |
| | *SNX29* | Sorting Nexin 29; circular RNA derived from this gene reduced myoblast proliferation and promoted cell differentiation |
| | *SLC16A1* | Solute Carrier Family 16 Member 1; a key controller of the cell cycle and mitosis, oncogene role in promoting cancer cell proliferation |
| | *MERTK* | MER Proto-Oncogene, Tyrosine Kinase; regulates cell survival, migration, differentiation, and phagocytosis of apoptotic cells; increases tumor immunogenicity |
| | *ARRDC1* | Arrestin Domain-Containing 1; a tumor suppressor in ccRCC in the Hippo pathway |
| | *STEAP1* | Six Transmembrane Epithelial Antigen Of The Prostate 1; promotes proliferation, migration, invasiveness, and tumorigenicity |
| STRAC14 | *ECE2* | Endothelin Converting Enzyme 2; involved in the processing of various neuroendocrine peptides |
| | *TMEM220* | Transmembrane Protein 220 |
| | *TOMM40* | Translocase of Outer Mitochondrial Membrane 40; essential for import of protein precursors into mitochondria |
| | *NDUFS6* | NADH:Ubiquinone Oxidoreductase Subunit S6; interact with CD147 in the mitochondria and may play a role in the multidrug resistance of cancer cells |
| | *NEB* | Nebulin; muscle protein that may be involved in maintaining the structural integrity of sarcomeres and the myofibril's membrane |
| | *ZNF727* | Zinc Finger Protein 727; may be involved in transcriptional regulation |
| | *IRF6* | Interferon Regulatory Factor 6; may be a transcriptional activator |
| | *ZNF483* | Zinc Finger Protein 483; may be involved in transcriptional regulation |
| | *ACHE* | Acetylcholinesterase (Cartwright Blood Group); has a role in neuronal apoptosis |
| | *LIPC* | Hepatic Triacylglycerol Lipase; plays a role in in cancer progression and metastasis |
| | *RAB3IL1* | RAB3A Interacting Protein Like 1; may have a role in synaptic vesicle exocytosis |
| | *SAMD15* | Sterile Alpha Motif Domain-Containing 15 |
| | *MTMR8* | Myotubularin Related Protein 8; in complex with MTMR9, negatively regulates autophagy |
| | *PKP1* | Plakophilin 1; may be involved in molecular recruitment and stabilization during desmosome formation |

*STRAC11, 13, 14* gene expression signatures of 11, 13, and 14 genes.

e). In the JAVELIN Renal 101 overall population, low versus high expression of STRAC11 GES was associated with longer PFS (HR 0.69 [95% CI: 0.56–0.85]) (Fig. 5f). In sunitinib-treated patients in the JAVELIN Renal 101 trial, low versus high expression of STRAC11 GES was associated with longer PFS (HR 0.58 [95% CI: 0.43–0.77]), but did not differentiate PFS in patients treated with avelumab plus axitinib (HR 0.90 [95% CI: 0.65–1.23]) (Fig. 5g).

We also investigated the placebo arm of the S-TRAC study to identify the molecular characteristics that might associate with a higher risk of recurrence, independent of anti-angiogenesis adjuvant therapy with sunitinib. Using the same approach, we identified a GES that consists of 14 genes (STRAC14) and that is set to be prognostic in nature, beyond the UISS clinic-pathological classification (Fig. S2A). In the overall study population, low versus high expression of STRAC14 GES was associated with longer DFS (HR 0.36 [95% CI: 0.21–0.64]) (Fig. S2B). In the placebo arm, low versus high expression of the STRAC14 GES was associated with longer DFS (HR 0.07 [95% CI: 0.02–0.24]) (Fig. S2C). Similar to STRAC11, the STRAC14 signature was also verified in the TCGA KIRC dataset and JAVELIN Renal 101 metastatic RCC population (Figs. S2D–G). Following the co-expression and

network analyses, the STRAC14 signature was also found to be enriched for genes involved in the regulation of the stroma component of the tumor, as well as Treg cells and myeloid cell subsets (Table 1).

Finally, we expanded this exploration with the overall sample set, looking for the group of patients at the highest risk of recurrence based on molecular criteria. Using the overall sample set, we identified a GES signature that consisted of 13 genes (STRAC13) (Fig. S3A). In the overall study population, low versus high expression of STRAC13 GES was associated with longer DFS in the overall study population (Fig. S3B) and in the sunitinib and placebo arms (Fig. S3C). The STRAC13 signature was also verified in the TCGA KIRC dataset and JAVELIN Renal 101 metastatic RCC population (Fig. S3D–G), and it was found to be enriched for genes involved in the regulation of the stroma component of the tumor as well as Treg cells, general T cells, and myeloid cell subsets (Table 1).

## Discussion

We report a comprehensive genomic and transcriptomic analysis of tumor samples from patients lacking overt metastases but at high risk for recurrent RCC post nephrectomy. These analyses were performed using data from patients treated in S-TRAC, a phase III trial comparing adjuvant sunitinib to placebo and meeting its primary endpoint of prolonging DFS. We confirmed the importance of the angiogenic context as a prognostic and predictive marker of the outcome. We identified three GES based on the sample set considered for the discovery. Each signature consists of genes representative of the tumor stroma microenvironment, including fibroblasts, Treg cells, and myeloid cells that, when highly expressed, were associated with poor outcomes and short DFS. The overlap in key elements captured in these three GES suggests that they are likely to be prognostic and also predictive of the anti-angiogenic effect in the adjuvant setting. These signatures also point to the identification of potential therapeutic targets to be considered for development in adjuvant RCC, such as Mer proto-oncogene tyrosine kinase (MERTK) and Tryptophan 2,3 dioxygenase (TDO2).

The STRAC11 GES (derived from the sunitinib treatment arm) identified a group of patients who are more likely to benefit from anti-angiogenic adjuvant therapy with sunitinib. When the published angiogenic signatures derived from studies in patients with metastatic RCC in IMmotion 150 GES and JAVELIN Renal 101 GES[18,20] were applied to the S-TRAC study population with localized RCC at high risk for recurrence, IMmotion 150 GES appeared to be prognostic while JAVELIN Renal 101 GES seemed to be predictive of outcomes with adjuvant sunitinib. While both signatures are pointing to a proangiogenic phenotype, the genes included in each signature are not overlapping; this is consistent with the fact that even though anti-angiogenic agents have been studied and approved in mRCC for more than a decade, there is still no clinically validated biomarker or signature used to identify the patients who will benefit from these agents; accordingly, additional work is required to further optimize the signatures reported to date. Thus, high levels of expression of angiogenesis signatures are associated with a better outcome, and treatment with anti-angiogenic therapy alone does not appear to increase the overall benefit, as these patients have more favorable risk factor characteristics regardless of treatment. In the sunitinib arm, high expression of the myeloid-inflamed phenotype was associated with poor outcomes. Altogether, these findings suggest that targeting the VEGF pathway alone may not be sufficient to prevent recurrence and that other factors such as immune response and the tumor environment are, at least, of similar importance. Checkpoint inhibitors are currently being studied in the adjuvant setting[41,42]. Recently, the Keynote-564 phase 3 trial demonstrated a benefit in relapse-free survival for pembrolizumab compared to placebo in the adjuvant setting of RCC following nephrectomy[43]. Use of the STRAC11 GES in the correlative analyses of the completed adjuvant trials studying single-agent

checkpoint inhibitors would further validate our findings and potentially help with better characterization of the patients at high risk of recurrence.

Furthermore, it is possible that the addition of a third agent that is able to modulate the tumor microenvironment may be an effective strategy to potentiate the response to the combined anti-angiogenic and PD-1/PD-L1 blockade in ccRCC. *MERTK* (part of STRAC11) encodes the Mer proto-oncogene tyrosine kinase (MERTK). Blockade of MERTK macrophage receptor increases tumor immunogenicity and potentiates antitumor immunity by inducing the tumor-cGAS- and host-STING-dependent type I interferon (IFN) response[44]. Activation of the c-GAS-STING pathway causes an increase in PD-L1 expression; however, when combined with PD-1 blockade, models of established tumors previously resistant to PD-1 blockade alone regressed[45].

*TDO2* (also part of STRAC11) is potentially another target for combination with anti-angiogenic therapy. Higher expression of *TDO2* was associated with poor survival outcomes in breast cancer[46,47]. Targeting *indolamine-2,3-dioxygenase-1* (*IDO1*) expression enhanced response to IFN treatment in RCC cell lines[48]. Furthermore, *IDO1* is known to promote tumor neovascularization by modulating the expression of IFN-γ and interleukin (IL)-6[49]. The immunomodulating activities of *IDO1* and *TDO2* led to the development of an inhibitor of these two enzymes, which is currently under investigation in a phase I trial (ClinicalTrials.gov, NCT03641794).

Other STRAC11 genes that have been implicated in kidney injury or cancer progression include *SLPI*[50,51], *SLC16A1*[52], and *ARRDC1*[53]. Downregulation of *STEAP1* increased proliferation and clonogenicity, and promoted cell migration, invasion, and the progress of mesenchymal transition in endometrial carcinomas[54] and breast cancer[55]. In gastric cancer, expression of *STEAP1* has been shown to promote proliferation, migration, invasiveness, and tumorigenicity[56].

On the DNA sequencing side, and in line with previous publications, we identified mutations in *MTOR*, which were slightly more frequent in the S-TRAC high-risk patient population than previously reported[27]. One recently reported phase 3 trial of adjuvant everolimus compared with surveillance alone following nephrectomy showed a numerical advantage in favor of everolimus, and a stronger signal in patients at very high risk for relapse[57]. Mutations in *MTOR* were enriched in S-TRAC patients selected to be at high risk for relapse were a poor prognostic marker. We also identified potential target genes that, when mutated, were associated with shorter DFS in the placebo arm, but prolonged DFS upon adjuvant treatment with sunitinib. *WDFY4* was reported to be essential for the cross-presentation of tumor-derived antigens and for antiviral and antitumor immunity[33]. Somatic mutations in *CSPG4* were found to be associated with membranous PD-L1 expression in Chinese patients with RCC[58]. *THEMIS* encodes a T-cell lineage-specific protein, which is essential for the maintenance of peripheral CD8+ T cells and for proliferative CD8+ T-cell responses to low-affinity peptide major histocompatibility complex signals aided by cytokines[30,31]. Interestingly, while high TMB was a poor prognosis in the placebo arm as previously established, it did not appear to have the same impact in the sunitinib treatment arm. This effect might be due to the fact that patients with high tumor CD8 positivity derived greater benefits from sunitinib treatment, as previously reported in this same cohort of patients[13].

Some limitations of the current analyses are due to the great majority of the patients who had disease recurrence with distant metastases, thus restricting our findings and conclusions to the metastatic process and not the local progression. In addition, due to the size of the subpopulation with available tumor tissue properly consented for these analyses (171 patients, 27.8% for WES; 133 patients, 21.6% for gene expression profiling) versus the total study population (615 patients), some of the analyses are likely to be insufficiently powered to detect potentially significant or meaningful differences.

In summary, our analysis suggests that the immunosuppressive environment of the micrometastasis at the nesting site is likely as important as, if not more than, the angiogenic phenotype in identifying effective therapy in the adjuvant RCC setting. It further suggests that adjuvant therapy should aim at targeting some of the key elements captured in the GES discovered in this study. Altogether, these findings may inform therapeutic strategies and more personalized approaches for adjuvant therapy in patients with RCC at high risk of recurrence.

## Methods

These analyses comply with all relevant ethical regulations. The S-TRAC trial was conducted in accordance with the ethics principles of the Declaration of Helsinki and Good Clinical Practice guidelines, defined by the International Council for Harmonization. All the patients provided written informed consent for inclusion in the trial and 212 patients provided informed consent for exploratory analyses. The protocol, amendments and informed consent forms were approved by the institutional review board or independent ethics committee at each trial site (Table S2). An independent external data monitoring committee reviewed efficacy and safety[11].

### Experimental model and subject details: S-TRAC study

In the S-TRAC trial[11], patients with RCC were randomized 1:1 to receive sunitinib (50 mg/day) or placebo on a 4-weeks-on/2-weeks-off schedule for nine cycles (-1 year), or until disease recurrence, the occurrence of secondary malignancy, significant toxicity, or consent withdrawal. Key inclusion criteria included: diagnosis with clear cell, loco-regional (defined as ≥T3 and/or N1-2) RCC; lack of macroscopic residual or metastatic disease confirmed by blinded independent central review (BICR); no prior systemic or anti-angiogenic treatment for RCC, and treatment initiation within 3–12 weeks of nephrectomy. The primary endpoint was the duration of DFS, defined as the interval between randomization and the first tumor recurrence, the occurrence of metastasis or a secondary cancer (as assessed by BICR), or death.

Of the 615 patients in the intent-to-treat population in S-TRAC trial, anonymized archival tumor tissue from nephrectomy or tumor biopsy (ormalin-fixed paraffin-embedded [FFPE] block or slides) were collected on an optional basis for only 212 patients who properly consented for exploratory analysis. Following the prior tumor tissue analyses, 193 individual specimens were available for molecular profiling, of which 171 (27.8%) (sunitinib, $n = 91$; placebo, $n = 80$) returned results for the WES analysis, and 133 (21.6%) (sunitinib, $n = 72$; placebo, $n = 61$) returned results for the GES analysis; 11 cases failed both DNA and RNA extractions, while another 11 failed the DNA extraction and/or did not pass QC (at the library or sequencing steps) and similarly 51 failed the RNA extraction and/or QC steps. This cohort represents a subset of the 191 patients previously included in the immunohistochemistry analysis for staining of PD-L1, CD4, CD8, and CD68[13]. It is also representing a subset of the 193 patients with T3 RCC included in the primary analysis of the 16-gene signature previously published[15]. Patient demographics and baseline characteristics were balanced between the sunitinib and placebo groups for the WES and GES analyses (Table S1). The study was conducted primarily in Europe, with fewer than 10% of patients enrolled in the United States and -10% enrolled in Asian countries, thus very few non-White patients were included in the study.

### Whole exome and transcriptome sequencing

Archival de-identified FFPE tumor tissue blocks from nephrectomy or tumor biopsy were obtained from patients who provided informed consent for genomic analyses. Sample were processed as per below and the same sections were used for both DNA and RNA extractions.

For WES (Accuracy and Content Enhanced [ACE] version 3; Illumina NovaSeq), resulting sequences were processed by the Personalis ACE Cancer Exome pipeline (Personalis, Inc, Menlo Park, CA), which uses BWA, GATK, MuTect, VarDict, and Picard to generate variant calls. Variant calls were further filtered using Personalis proxy-normal database (which consists of in part the use of normal variant database MuTect, the Genome Aggregation Database (gnomAD; https://gnomad.broadinstitute.org)[59] and custom filters to remove many germline variants found in normal tissue as well as sequencing platform bias. Mutations with a minimum of five mutant reads (that is, found on at least five separate DNA molecules in an individual tumor sample) that were not annotated as synonymous variants and annotated as resulting in a change in protein coding sequence were included in the analysis.

The clinical significance of TMB was assessed through WES analysis. The presence of non-synonymous single nucleotide variants per megabase (NSSNV/Mb) defined the TMB high and low status relative to the median value.

Whole-transcriptome profiles were generated using RNA-seq (ACE version 3) on FFPE tumor tissue. Transcript levels were quantified by the Personalis ACE Cancer Transcriptome Analysis pipeline (Personalis, Inc, Menlo Park, CA), which uses STAR version 2.4.2a-p1 to align reads to the NCBI hs37d5 annotation 105 reference genome and produces transcripts per million (TPM) values for each gene. TPM values were log2 transformed for further analysis of individual genes or standardized gene pathway signature scores. Briefly, for each gene, we calculated the mean expression and s.d. across samples. Then, we subtracted the mean and divided by the s.d. to standardize the gene score to be centered at zero with units of s.d. (z-score). The pathway score for each sample was calculated as the average of the standardized values for the set of genes within the pathway. To assess sample purity and rule out potential bias, tumor purity was calculated from the corresponding RNA-seq expression data using the ESTIMATE method[60]. Using the RNA ESTIMATE Tumor Purity score, the tumor purity ranged from 28% to 92% with a median of 55%. Overall, tumor purity did not bias the tumor mutation burden; the mutant status of individual genes was not associated with increased tumor purity; and it did not confound gene expression.

### Gene expression signature analyses

GES analyses included published signatures from the IMmotion 150 study "[Teff, angiogenesis, myeloid inflammation (Minf)]", and the JAVELIN Renal 101 study [Immune, angiogenesis][18,20].

TCGA subtype classification labels by unsupervised gene expression clustering[27] were used to define centroids from the z-scored scaled expression level of the top 20% variable genes from the TCGA KIRC dataset. Expression data from the trial were z-scored and mapped to the nearest centroid to assign the subtype label. The method used was implemented by the Data4Cure molecular affinity app. RNA-seq data was processed using xCell to generate scores for 64 cell type signatures[38].

**Elastic net analysis.** Multi-feature signatures were derived using samples with complete data from the sunitinib arm for STRAC11, the placebo arm for STRAC14, and from both arms for STRAC13. For each dataset, we performed 1000 bootstrap runs of fitting a Cox proportional hazards model for DFS with regularization by an elastic net penalty[61] and a fivefold cross-validation. Features were ranked by the frequency observed in the bootstraps, and the number of top features was selected using a local maximum concordance index. A composite signature score was computed by a weighted sum of the top features, and each feature was weighted by its average coefficient across bootstrap models. The STRAC14, STRAC13, and STRAC11 GES were further verified using two independent datasets, the JAVELIN Renal

101 study in patients with metastatic advanced RCC (*n* = 720) and the KIRC dataset of TCGA (*n* = 488).

### Quantification and statistical analysis

For gene expression analyses, DFS by BICR was compared between biomarker stratum by <median vs ≥median values of a particular parameter using Kaplan–Meier analysis. DFS was also compared between the two treatment groups within a biomarker stratum using the Kaplan–Meier method. For the mutational analyses, Cox proportional hazards model with wild-type samples as the reference group was used to calculate HR and 95% CI. A HR <1 indicates better DFS in the mutant group, whereas a HR >1 indicates better DFS in the wild-type group. Log-rank two-sided test was performed to compare between wild-type and mutant groups. Multivariate Cox proportional analysis of DFS by BICR was performed on the gene expression and mutational data adjusted by the patient's age and sex. No adjustments of *p*-values or CIs for multiplicity were performed.

### Reporting summary

Further information on research design is available in the Nature Research Reporting Summary linked to this article.

## Data availability

Relevant summarized data and supplemental materials cited in the manuscript are publicly available[62]. Source data are available with this manuscript. The tumor tissue sequencing raw data that support the findings of this study have been submitted to the European Genome-Phenome Archive (EGA) with accession numbers EGAS00001006528 (transcriptome sequencing) and EGAS00001006529 (exome sequencing); due to informed consent limitations on patient confidentiality and secondary use of data, restrictions apply to the availability of the data. Anonymized data can be made available from the authors under a Data Access Agreement upon reasonable request and with permission of the Pfizer Data Access Committee.

Additional datasets used in this analysis included subsets of TCGA databases (https://doi.org/10.1038/nature12222; https://xenabrowser.net/datapages/?dataset=tcga_RSEM_gene_tpm&host=https://toil.xenahubs.net; https://portal.gdc.cancer.gov/projects/TCGA-KIRC) and a subset of data from the JAVELIN Renal 101 Study B9991003 (https://www.nature.com/articles/s41591-020-1044-8#Sec33; https://static-content.springer.com/esm/art%3A10.1038%2Fs41591-020-1044-8/MediaObjects/41591_2020_1044_MOESM3_ESM.xlsx; Clinical Table S11 and Expression Table S13). Source data are provided with this paper.

## Code availability

The R scripts used for elastic net analyses are available[62]. R statistical software environment version 3.5 was utilized (https://www.r-project.org/). Data4Cure was utilized for analysis and visualization (https://www.data4cure.com). SAS version 9.4 (TS1M5) was utilized for data analysis (https://www.sas.com/). Processing of Whole Exome and Whole Transcriptome raw sequencing data into summarized data was performed at Personalis (https://www.personalis.com/biomarker-discovery/).

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

## Acknowledgements

The authors thank the participating patients and their families, as well as the investigators, sub-investigators, research nurses, study coordinators, and operations staff. The authors also thank Amber Donahue, PhD, for support with the biomarker analyses for this study. This study was sponsored by Pfizer. Patients treated at Memorial Sloan Kettering Cancer Center were supported in part by Memorial Sloan Kettering Cancer Center Support Grant/Core Grant (P30 CA008748). Medical writing support was provided by Vardit Dror, PhD, and Leon Adams, PhD, of Engage Scientific Solution and was funded by Pfizer.

## Author contributions

Conceptualization and Methodology: R.J.M., A.R., and J.-F.M. Acquisition of data: R.J.M., A.R., D.J.G., H.S.P., M.S., and O.V. Analysis and interpretation of data: All authors. Writing–original draft: All authors. Writing—review & editing: All authors. Formal analysis: X.L., X.J.M., and K.A.C. Funding acquisition: J.-F.M. Supervision: J.-F.M.

## Competing interests

R.J.M. receives consulting fees from Pfizer, Eisai, Novartis, Merck, Genentech/Roche, Incyte, Lilly Oncology, EMD Serono, Aveo, and Exelixis; and clinical trial support to his institution from Eisai, Bristol Myers Squibb, Exelixis, Genentech/Roche, Merck, Novartis, and Pfizer. M.S. receives grant support and fees for serving on advisory boards for Pfizer, GlaxoSmithKline, Novartis, Bayer, Exelixis, and Roche; and consulting fees, honoraria, and travel support from Pfizer, GlaxoSmithKline, Novartis, Bayer, and Roche. D.J.G. receives honoraria and consulting fees from Sanofi, Exelixis, and Bayer; consulting fees from Merck and Sanofi; grants from Genentech/Roche, Novartis, Astellas, Celldex, and Acerta; and grants and consulting fees from Exelixis, Janssen, Pfizer, Innocrin Pharma, and Bristol Myers Squibb. H.S.P. has received honoraria for advisory work from Ipsen and Eisai. J.-F.M., X.J.M., O.V., X.L., and K.A.C. are employees of and own stock in Pfizer. A.R. receives fees for serving on advisory boards from Pfizer, Novartis, GlaxoSmithKline, Bristol Myers Squibb, AstraZeneca, Merck GA, Merck Sharp & Dohme, Ipsen, and Roche; lecture fees from Pfizer, Novartis, Ipsen, and GlaxoSmithKline; travel support from Pfizer, Novartis, GlaxoSmithKline, Bristol Myers Squibb, Ipsen, Merck GA, and Merck Sharp & Dohme; and grant support from Pfizer, Merck GA, and Novartis.
