## [Peer review file · Nature Communications]

REVIEWER COMMENTS

Reviewer #1 (Remarks to the Author): Expert in renal cell carcinoma genomics and therapy

This study investigates the very important question of the use of adjuvant therapy in the treatment of patients with a high risk for recurrent disease post nephrectomy and whether biomarkers can be identified for the patients who would receive the greatest benefit. This study shows that the majority of the known frequency somatically altered ccRCC cancer genes did not influence recurrence with the exception of MTOR and ARID1A. The study also identified variations in several other novel genes that influenced the effectiveness of adjuvant therapy. Importantly, the study identified several gene expression signatures that were capable of delineating between patients more likely to recur and patients more likely to be progression free. These GESs were confirmed in two other publicly available ccRCC cohorts. The STRAC11 GES also showed evidence of being able to identify patients that would benefit the most from adjuvant therapy and this can be confirmed by future studies. These GESs also highlighted the importance that immune response and tumor environment have in influencing recurrence and response to VEGFR-targeted adjuvant therapy. This is consistent with the suggestion that the authors put forward that VEGFR-targeted adjuvant therapy alone may be sufficient and that a combinatorial approach may be more successful. This may also explain the lack of response to adjuvant therapy seen in other studies.

The discovery of new mutations in immune-related or chromatin homeostasis genes that influenced treatment outcome is very important. These genes had very high mutation frequencies and it is unclear from the text whether these changes are somatic or germline? Were the same mutations occurring in multiple patients? A list of the identified mutations could be beneficial. The methods state that only tumor tissues were sequenced and without associated normal sequencing the mutations seen in the tumors could be either somatic or germline. Looking on cBioPortal, these genes show a very low rate of somatic mutation in kidney cancer and this seems to suggest that germline mutation is more likely. It would be very important if these markers were in the germline DNA of these patients as screening for them could be easier.

The study is well written and provides significant and novel insight into a very important issue

I have several other minor comments listed below:

1. Introduction - Line 53, it would be useful to know how many clinical trials did not show clinical benefit and whether there was a reason that S-TRAC did? Was it a large trial or had longer follow-up? Was high risk always judged by the same criteria?

2. Tumor mutation burden can be calculated either with or without associated normal tissue sequencing. Can the authors clarify which methodology was used?

3. While it clearly stated in the methods, it would be beneficial to note in the results section that the transcriptome signature analysis was performed on only 133 of the 171 tumors for which the sequencing analysis was performed.

4. The STRAC11 GES was associated with longer PFS and OS in the TCGA KIRC patients but only associated with longer PFS in the JAVELIN Renal 101 patients. Was there no significant association with overall survival in the JAVELIN Renal 101 patients or was the data not available? This is true for the STRAC13 and STRAC14 analysis as well.

5. While both the IMmotion 150 and JAVELIN renal 101 angiogenesis signature showed an association with disease free survival in the study population, the JAVELIN signature was considerably more convincing and significant. The authors note the lack of overlap between the genes in these two signatures and it would be worth emphasizing in the discussion that not all angiogenesis signatures are the same and that some optimization of this signature could be beneficial.

6. The AMT gene was a component of both the STRAC11 and STRAC13 gene expression signatures. It might be beneficial to comment on this gene in the discussion. The human cancer protein atlas identifies AMT as a favorable prognostic marker in renal cancer (<https://www.proteinatlas.org/ENSG00000145020-AMT/pathology>). It is involved in the 1-carbon metabolic pathways and is encoded on chromosome 3p21.31 (where SETD2 is also located) where it is likely to have had single copy loss in most ccRCC tumors.

Reviewer #2 (Remarks to the Author): Expert in cancer genomics, gene signatures, statistics, and biomarkers

This is a very interesting study, which utilizes data from S-TRAC clinical trial. New WES was performed for 171 patients. Elastic net was explored to identify a potential signature to classify patients based on their favorable or poor response to treatment.

To enhance clinical applicability of this study, I would propose addressing the following points:

Major comments:

1. Why only 171 patients were used in this manuscript. The clinical trial page lists 674 participants (not clear why this specific subset was selected). Also, it is not clear why a different number of patients was used in citation 13 (by the same senior author) – are 171 patients a subset of the patients described in reference 13?
2. What was the sample purity (tumor content) of submitted 171 patients – for whole exome sequencing? How could it possibly affect the results? Could this tumor purity somewhat affect KM analyses (for that tumor purity needs to be estimated for the WES samples).
3. Does Suppl Fig 1 imply that tumors samples were not pure enough (was this done using RNA-seq profiles?) – this should be discussed and addressed. Association between sample purity and outcomes should also be assessed, alongside other variables (to ensure that these variables are not simply accidentally correlated with sample purity)
4. It is not clear why sunitinib-treated group is marked as black on some plots and as gray in other plots – it could create confusion (see Figure 1 vs Figure 2; etc) and is perhaps related to point 5 below.
5. It is not clear how conclusions on line 135-136 were reached. It is counter-intuitive using just the data provided and the way this figure is labeled. To achieve these conclusions, more detailed statistical analysis is needed.
6. 16-gene recurrence score should be evaluated in the 171 patients (as it was evaluated in a different/larger patient population in reference 13).
7. It is not clear why figure 3 only shows 131 patients. Why a cohort with 171 patients now used? This should be carefully discussed.
8. It is not clear what data type was used for elastic net – transcriptomic or genomic (WES?) Why was a specific data type used? Was the other data type considered? Results from both transcriptomic and genomic analyses should be evaluated and compared.
9. For elastic net, was the analysis adjusted for clinical variables: age, gender, initial stage (or anything else that is relevant) etc? Such adjustment should be performed.
10. Markers that are also prognostic in placebo arm should be excluded from the “11-gene” treatment-response signature – as they correspond to markers of overall disease aggressiveness and are not specific to treatment response (TCGA dataset supports this point). In other words, only markers specific to treatment response should be included in the final “11-gene signature”. One idea: perhaps these 11 markers can be used as input variables in the elastic net, with placebo arm only. Markers with significant prognostic importance in the placebo arm should be eliminated from the treatment-response “11-gene signature” (which will now have fewer genes).

Minor comments:

1. In the abstract: colon should be replaced with a comma and an “and”.

2. I would be careful about conclusions about “trends” if CI crosses 1.
3. I would suggest matching the exact signature names from Fig 3b to the KMs on the same Figure (including their prefixes).

Reviewer #3 (Remarks to the Author): Expert in renal cell carcinoma clinical research and sunitinib therapy

In this paper, Dr Motzer and coll. report the results of an integrated multi-omics tumor analysis of 171 patients treated with sunitinib or placebo within the S-TRAC trial, which tested the adjuvant role of sunitinib in high-risk patients following nephrectomy, in order to identify new molecular subtypes associated to these patients and potential new targets.

The study identified new gene expression signatures (GES), including the one consisting of 11 genes defined STRAC11 (derived from the sunitinib-treated population). It also investigated and tested the predictive/prognostic role of gene panels established in the advanced metastatic setting studies IMmotion 150 and Javelin RCC 101, in which sunitinib was used as a comparator treatment, since at the time of the study it was a standard of care for advanced/metastatic stage disease.

The results confirm the landscape of mutations already reported by the TCGA, with VHL, PBRM1, SETD2 and BAP1 as the most frequently mutated in localized RCC. Interestingly, this population resulted enriched in mTOR mutations, implicated in the control of angiogenesis, and that correlated with poor prognosis.

New mutations in THEMIS, WDFY4 or CSPG4 (genes related to T-cell activation, maturation and proliferation), were found to correlate with longer DFS in sunitinib-treated patients. In this regard, it would be interesting to know whether there are corresponding differences in the expression of these genes in patients receiving placebo compared to sunitinib, if feasible, to better distinguish between their (eventual) prognostic and/or predictive impact.

The authors also found that high tumor mutational burden (TMB) correlates with a better outcome only in patients receiving the placebo and not in those treated with sunitinib. It could be of interest to explain if this evidence is the effect of sunitinib (which is able to abrogate such predictive effect of TMB) or it is an epiphenomenon related to a confounding variable, such as a greater activation of genes related to T-cells activation and maturation in the sunitinib arm, as they previously reported. Again, the possible prognostic implication of TMB should be extensively discussed by the authors.

Authors report that two signatures (IMmotion 150 and Javelin Renal 101) are both related to different prognosis when used in the S-TRAC population, however, they do not explain why the significance was lost when the analysis was conducted only in placebo or sunitinib treated patients (as shown in figure 3b). A comment should be added.

Along with STRAC-11, authors report two other gene signatures defined STRAC13 and STRAC14, which were also “validated” in the TCGA-KIRC and in the JAVELIN Renal 101. Authors found that STRAC-11 did not differentiate DFS in the axitinib-avelumab treated patients within the Javelin Renal 101 population; the same conclusions are true also for STRAC14 and STRAC13. This paragraph is a bit confusing and it is difficult to understand the ranking among the GES and the reasons to prefer STRAC11 compared to STRAC13 or STRAC14. Authors are invited to clarify this part, reporting pros and cons of each panel and moving the less relevant results to Supplementary material.

The authors have also found and proposed MERTK and TDO2 as two potential new therapeutic targets in RCC. This is very preliminary and speculative conclusion at this stage and it would be important to analyse and describe also their prognostic role.

In the Discussion, authors should better explain the limits of the panels used (IMmotion 150 and Javelin Renal 101), if they have been previously studied together and what are the results. It is also important to specify that cut-offs are always different because they are based on median values of the studied populations.

Finally, this paper should be put in the context of immunotherapy, considering the classification used for patients eligible for adjuvant therapy in the Keynote-564 trial. It could be of interest to evaluate whether the incidence of mutations is different, according to risk-class populations.

It is not appropriate the definition “a trend”, when reporting results near the value of $p < 0.05$.

Although interesting and stimulating, the findings of this study are mainly descriptive of the landscapes of genes altered in the study population, and do not provide clues for a direct translation into practice. The same authors honestly underline other limits of the study, including: 1) the relative role of angiogenic signature and phenotype, demonstrated by the fact that adjuvant therapy with sunitinib does not prevent metastasis, and the greater relevance of tumor environment and immune response. 2) the fact that the great majority of patients who recurred had distant metastases, limiting the possibility to draw conclusions on the local progression; 3) the low percent of patients for whom adequate tumor tissue was available (only 27.8% of the total study population), which greatly limits the power of the analysis.

In addition, probably due to the need to comply with the constraints of words for publication in Nat Comm, the text is very often difficult to read with too many data from different studies and data set, finally resulting a bit confounding.

Reviewer Comments

Reviewer #1 (Remarks to the Author): Expert in renal cell carcinoma genomics and therapy

This study investigates the very important question of the use of adjuvant therapy in the treatment of patients with a high risk for recurrent disease post nephrectomy and whether biomarkers can be identified for the patients who would receive the greatest benefit. This study shows that the majority of the known frequency somatically altered ccRCC cancer genes did not influence recurrence with the exception of MTOR and ARID1A. The study also identified variations in several other novel genes that influenced the effectiveness of adjuvant therapy. Importantly, the study identified several gene expression signatures that were capable of delineating between patients more likely to recur and patients more likely to be progression free. These GESs were confirmed in two other publicly available ccRCC cohorts. The STRAC11 GES also showed evidence of being able to identify patients that would benefit the most from adjuvant therapy and this can be confirmed by future studies. These GESs also highlighted the importance that immune response and tumor environment have in influencing recurrence and response to VEGFR-targeted adjuvant therapy. This is consistent with the suggestion that the authors put forward that VEGFR-targeted adjuvant therapy alone may be sufficient and that a combinatorial approach may be more successful. This may also explain the lack of response to

adjuvant therapy seen in other studies.

The discovery of new mutations in immune-related or chromatin homeostasis genes that influenced treatment outcome is very important. These genes had very high mutation frequencies and it is unclear from the text whether these changes are somatic or germline?

Response: We thank the reviewer for this observation and associated question. To clarify the way the tumor tissue was analyzed for WES, in absence of normal control sequencing, additional information has been included in the method. In particular, variant calls were further filtered using Personalis proxy-normal database (which consists of in part the use of normal variant database Mutect, the Genome Aggregation Database (gnomAD; <https://gnomad.broadinstitute.org>) and custom filters to remove many germline variants found in normal tissue as well as sequencing platform bias. However, this filtering is not always efficient or successful. We performed an additional round of germline variant filtering to address this point. Variants were first annotated using myvariant.info; If a dbsnp entry existed for the specific variant allele, then we used dbgap_popfreq, or gnomAD if not available, for Minor Allele Frequency value. If there was no dbsnp entry for the specific variant allele, then the variant was retained. If the Minor Allele Frequency value was $> 2\%$, then it was marked as germline and filtered out. Following this additional process, most variants were actually very rare. For the following genes, no changes were observed: THEMIS, CSPG4, KMT2D, PPIP6K1; or limited changes were noted (but direction and magnitude of effect conserved or improved): CTCFL and WDFY4. The only gene for which the germline subtraction had an effect was ERICH6B since the variant pS174T was the main variant detected, and it is

germline. We have clarified accordingly in the Results section with the following statement: “of note, the main variant allele detected for ERICH6B, pS174T, is actually a germline variant.”

Were the same mutations occurring in multiple patients? A list of the identified mutations could be beneficial.

Response: In general, the mutations observed were different in each patient except for the variant allele noted above (ERICH6B, pS174T), which is actually a germline variant and for THEMIS where a substitution in position 613 was observed (p.H613Q, c.1839C>A; p.H613R, c.1838A>G; or p.H613R, c.1838_1839delACinsGA).

The methods state that only tumor tissues were sequenced and without associated normal sequencing the mutations seen in the tumors could be either somatic or germline. Looking on cBioPortal, these genes show a very low rate of somatic mutation in kidney cancer and this seems to suggest that germline mutation is more likely. It would be very important if these markers were in the germline DNA of these patients as screening for them could be easier.

The study is well written and provides significant and novel insight into a very important issue

I have several other minor comments listed below:

1. Introduction - Line 53, it would be useful to know how many clinical trials did not show clinical benefit and whether there was a reason that S-TRAC did? Was it a large trial or had longer follow-up? Was high risk always judged by the same criteria?

Response: We thank the reviewer for this suggestion, and we have added the following with the associated references:

“There are various potential reasons for which S-TRAC had a positive outcome while the other trials did not, including the overall drug exposure and the clinicopathologic selection of the patient population enrolled (Figlin et al Ann Oncol. 2018 ; Lenis et al J Urol. 2018 ; Salmasi et al European Urology 2018). ”

However, we believe a full discussion of the discrepant study results is beyond the scope of this manuscript.

2. Tumor mutation burden can be calculated either with or without associated normal tissue sequencing. Can the authors clarify which methodology was used?

Response: Briefly, the TMB was calculated without the associated normal tissue sequencing subtraction and was based on the non-synonymous single nucleotide variants. To clarify these points, and other, around the methodology utilized for the WES and whole transcriptome analyses, a new section has been added in the Methods under the “Whole Exome and Transcriptome Sequencing” section.

3. While it clearly stated in the methods, it would be beneficial to note in the results section that the transcriptome signature analysis was performed on only 133 of the 171 tumors for which the sequencing analysis was performed.

Response: We recognize that this information is missing in the Results section. We have added the following:

“Next, we investigated previously defined gene expression signatures (GES) established in the advanced metastatic setting (Stage IV) in the two studies, IMmotion 150 and JAVELIN Renal 101 in the subset of 133 cases that were successfully profiled in the STRAC trial.”

4. The STRAC11 GES was associated with longer PFS and OS in the TCGA KIRC patients but only associated with longer PFS in the JAVELIN Renal 101 patients. Was there no significant association with overall survival in the JAVELIN Renal 101 patients or was the data not available? This is true for the STRAC13 and STRAC14 analysis as well.

Response: At the time of the analysis, and as most recently published (Choueiri et al, *Annal Oncol.* 2020. 31(8), 1030-1039), the JAVELIN Renal 101 OS data were still immature. Thus, associations between the STRAC signatures and the JAVELIN Renal 101 OS data were not tested.

5. While both the IMmotion 150 and JAVELIN renal 101 angiogenesis signature showed an association with disease free survival in the study population, the JAVELIN signature was considerably more convincing and significant. The authors note the lack of overlap between the genes in these two signatures and it would be worth emphasizing in the discussion that not all angiogenesis signatures are the same and that some optimization of this signature could be beneficial.

Response: The authors thank the reviewer for pointing this important information. The following sentence has been added in the discussion: “Importantly, while both signatures

are pointing to a proangiogenic phenotype, the genes included in each signature are not overlapping; this is consistent with the fact that even though anti-angiogenic agents have been studied and approved in mRCC for more than a decade, there is still no clinically validated biomarker or signature used to identify the patients who will benefit from these agents; accordingly, additional work is required to further optimize the signatures reported to date.”

6. The AMT gene was a component of both the STRAC11 and STRAC13 gene expression signatures. It might be beneficial to comment on this gene in the discussion. The human cancer protein atlas identifies AMT as a favorable prognostic marker in renal cancer (<https://www.proteinatlas.org/ENSG00000145020-AMT/pathology>). It is involved in the 1-carbon metabolic pathways and is encoded on chromosome 3p21.31 (where SETD2 is also located) where it is likely to have had single copy loss in most ccRCC tumors.

Response: The AMT gene was found to be on both STRAC11 and STRAC13 gene expression signatures, and was the only gene represented in more than 1 of the 3 signatures. According to the human cancer protein atlas project, AMT expression is ubiquitous across cancer type and seems to be prognostic in RCC, with high AMT expression being a favorable factor, regardless of the cancer stage. This enzyme is critical to the metabolism of glycine and is potentially enriched in NK-cells. However, limited information is published for this gene and its role in RCC and the authors would prefer not include this new item into the discussion.

Reviewer #2 (Remarks to the Author): Expert in cancer genomics, gene signatures, statistics, and biomarkers

This is a very interesting study, which utilizes data from S-TRAC clinical trial. New WES was performed for 171 patients. Elastic net was explored to identify a potential signature to classify patients based on their favorable or poor response to treatment. To enhance clinical applicability of this study, I would propose addressing the following points:

Major comments:

1. Why only 171 patients were used in this manuscript. The clinical trial page lists 674 participants (not clear why this specific subset was selected). Also, it is not clear why a different number of patients was used in citation 13 (by the same senior author) – are 171 patients a subset of the patients described in reference 13?

Response: Of the 610 patients who received either sunitinib or placebo treatment (615 patients were enrolled in the study), only 212 had tumor samples (FFPE block or slides) available; these were collected on an optional basis and had to be properly consented for molecular profiling prior to be anonymized for use. These samples were previously analyzed for immune marker expression by IHC (196 tissue blocks submitted but only 191 cases with sufficient tumor tissue content; reference #11) and for the 16-gene expression signature (212 samples submitted, including 193 T3 RCC for the primary analysis; reference #13).

We then submitted the leftover tumor specimens to Personalis for WES/WT analyses. There were 193 patients' specimens sent for analysis; following DNA and RNA extraction, 11 cases fail both extractions, while another 11 fail the DNA extraction and/or did not pass QC (at the library or sequencing steps) and similarly 51 fail the RNA extraction and/or QC. In summary, 171 cases returned WES results and were included in the analysis, while 133 cases returned results and were including in the GES analyses. And all of these were included in the 16-gene signature manuscript; because of the technique utilized for the 16-gene signature (qRT-PCR) which is more forgiving in term of RNA quality and quantity, more samples could be included in the analysis (including samples that had tumor tissue slides only) vs. what could be done for the RNAseq whole transcriptome analysis. More specifically, out of the 193 T3 cases included in the reference #13, 157 (81.3%) were included in the current WES population, and 125 (64.8%) in the GES population.

We have provided more information in the Methods to capture these important elements.

2. What was the sample purity (tumor content) of submitted 171 patients – for whole exome sequencing? How could it possible affect the results? Could this tumor purity somewhat affect KM analyses (for that tumor purity needs to be estimated for the WES samples).

Response: By whole slide H&E, the tumor cell content of the specimen utilized varied from 5% to 100%, with the median situated above 60%. This estimate does not take into account any macro dissection steps that might have been performed prior to the extraction.

Hence to get a better estimation of the role of the sample purity, tumor purity was calculated from the corresponding RNA-Seq expression data using the ESTIMATE method. (Yoshihara, et al. Nat Commun. 2013;4:2612). Since the same slide tissue was digested and separated into RNA and DNA, the RNA comes from the same cells as the DNA. Using the RNA ESTIMATE TumorPurity score, the tumor purity ranged from 28% to 92% with a median of 55%.

We tested the association between the RNA ESTIMATE of tumor purity, DFS, and mutation status. Tumor purity itself was not associated with DFS in either arm or in the whole cohort. There was no significant difference (*t*-test) in tumor purity between wildtype and mutant samples for each of the genes of interest (THEMIS, WDFY4, CSPG4, CTCFL, KMT2D, PPIP5K1). There was a weak Pearson correlation between TMB and tumor purity due to several samples with very high TMB. But in the majority of samples, TMB was independent of tumor purity. Thus overall, tumor purity did not bias the tumor mutation burden and the mutant status of individual genes was not associated with increased tumor purity. This information has been added in the Methods section.

3. Does Suppl Fig 1 imply that tumors samples were not pure enough (was this done using RNA-seq profiles?) – this should be discussed and addressed. Association between sample purity and outcomes should also be assessed, alongside other variables (to ensure that these variables are not simply accidentally correlated with sample purity)

Response: It is true that the Fibroblast gene expression signature is loosely correlated with tumor purity metrics. Indeed, the ESTIMATE method uses a stromal expression score in the calculation of tumor purity. The distribution of tumor purity (*t*-test) was not significantly different between the placebo and sunitinib arms ($p=0.6$) and tumor purity was not significantly associated with DFS in either arm. Yet, the association of the fibroblast signature was very specific for the sunitinib arm only and not the placebo arm. Thus, the observation is not confounded by tumor purity.

4. It is not clear why sunitinib-treated group is marked as black on some plots and as gray in other plots – it could create confusion (see Figure 1 vs Figure 2; etc) and is perhaps related to point 5 below.

Response: We thank the reviewer for pointing these discrepancies. Accordingly, the Figure 2 B was corrected so that Sunitinib figures in gray, while Placebo figures in black. For consistency, the Figure 5 G was also modified to keep Sunitinib in gray. Lastly, Figure 3B was corrected by replacing “Favors Mutation” and “Favors Wild type” for the signatures, by “Favors \geq Median” and “Favors $<$ Median”.

5. It is not clear how conclusions on line 135-136 were reached. It is counter-intuitive using just the data provided and the way this figure is labeled. To achieve these conclusions, more detailed statistical analysis is needed.

Response: The authors acknowledge that there might have been some confusion around the color scheme of Figure 2B, and also the fact that the same color scheme is used on figure 2A but does not refer to the same groups. The authors also provided additional information around the TMB calculation in the Method section (please refer to Reviewer #1, item #2). The authors also recognize that the data are limited to draw the strong conclusions stated on line 135-136 and have modified the sentence accordingly as follows:

“Interestingly, treatment with sunitinib in the adjuvant setting largely abrogated this prognostic effect (HR 0.76 [95% CI: 0.37–1.55]; $p = 0.5619$; Figure 2B), suggesting that high TMB may be confounded with ~~may be predictive~~ of a clinical benefit to adjuvant sunitinib.”

6. 16-gene recurrence score should be evaluated in the 171 patients (as it was evaluated in a different/larger patient population in reference 13).

Response: As mentioned in point #1, these 171 patients are a subgroup of the patients included in the 16-gene signature analysis, and more specifically 157 (81.3%) stage T3 cases reported in the reference #13. We have clarified this point in the Method.

7. It is not clear why figure 3 only shows 131 patients. Why a cohort with 171 patients now used? This should be carefully discussed.

Response: As mentioned in points #1, there were 193 patients' specimens sent for analysis at Personalis for WES and WTS; following DNA and RNA extraction, 11 cases fail both extractions, while another 11 fail the DNA extraction and/or did not pass QC (at the library or sequencing steps) and similarly 51 fail the RNA extraction and/or QC. In summary, 171 cases returned WES results and were included in the analysis, while 133 cases returned results and were including in the GES analyses. We have updated the Methods section as well as clarify these points in the Results section.

8. It is not clear what data type was used for elastic net – transcriptomic or genomic (WES?) Why was a specific data type used? Was the other data type considered? Results from both transcriptomic and genomic analyses should be evaluated and compared.

Response: The authors thank the reviewer for the question. Transcriptomic data from RNAseq on the tumor samples were used for the elastic net model, as a follow up to and to be consistent with the previous signatures previously discovered and reported for metastatic RCC (references 16 and 18). We have clarified this in the revised manuscript. We have taken the reviewer's suggestion and constructed another elastic net model taking into account both transcriptomic and WES genomics data as well as clinical variables including age and gender. The initial stage was not included in this model since as stated in the response to item # 6 and as per the Table S1 in the manuscript, over 91% of patients included in this analysis were Stage T3. The elastic net regression identified a model consisting of 21 features including 17 gene expression features and 4 mutation

features, while clinical variables were not selected as top features that went into the model. The multivariate signature derived from the 21-feature model is highly predictive (c-index=0.91; By median multivariate signature cutoff in the sunitinib arm, signature high vs. low HR=19 and p-val= 1.5×10^{-7} with comparable performance to the STRAC11 gene expression signature (c-index=0.87; By median multivariate signature cutoff in the sunitinib arm, signature high vs. low HR=22 and p-val= 6.3×10^{-8}). This suggest that, by adding genomics data and clinical parameters, the model performance, as measured by the c-index, only increases slightly. Therefore, and to keep consistent with the type of data and signatures previously published in references #16 and 18, the additional model was not included into the revised manuscript.

Briefly, the model was constructed as follow:

- Filter out lowly expressed and low variable genes, leaving 11,290 genes
- Filter out non-coding genes, leaving 9,423 genes for expression
- WES data filter for frequency >5% in the cohort, leaving 2,465 genes with mutation data
- Also include age and gender in the modeling
- 70 samples with complete data and are in the sunitinib arm
- Run 5-fold cross-validation on the sunitinib arm
- Run bootstrapping on the samples for 1,000 times and calculate the average coefficients and bootstrapping frequencies of the genes.

This resulted in the following 21-feature elastic net signature:

Bootstrapping Frequency

9. For elastic net, was the analysis adjusted for clinical variables: age, gender, initial stage (or anything else that is relevant) etc? Such adjustment should be performed.

Response: The models presented in the manuscript were not adjusted for the suggested clinical variables. However, as mentioned in the response to item #8, the impact of the initial stage was limited, if any, since >90% of the patients included in the current analysis were Stage T3. As per the Supplemental Table 1, there were only 3 patients, T4 N0 or NX, M0, and there are 11 patients, Any T, N1-2, M0 included in the analysis. Hence the adjustment considered for inclusion in the new Elastic Net model proposed in item #8 included age and gender only but were not selected as top features that went into the model.

10. Markers that are also prognostic in placebo arm should be excluded from the “11-gene” treatment-response signature – as they correspond to markers of overall disease aggressiveness and are not specific to treatment response (TCGA dataset supports this point). In other words, only markers specific to treatment response should be included in the final “11-gene signature”. One idea: perhaps these 11 markers can be used as input variables in the elastic net, with placebo arm only. Markers with significant prognostic importance in the placebo arm should be eliminated from the treatment-response “11-gene signature” (which will now have fewer genes).

Response: the authors thank the reviewer for this comment. While we recognize that each of the 11-genes in the STRAC11 signature individually may have either predictive or prognostic value in univariate analysis and that, as the reviewer pointed out, some feature may also be prognostic in the placebo arm, the approach followed, to derive a multivariate feature model using an elastic net regression framework, is an unbiased unsupervised approach to automatically select multiple molecular features which, in aggregate, are the most predictive of sunitinib outcome.

Some features in STRAC11, while individually, may also be prognostic in the placebo arm and did not meet the statistical significance to be ‘predictive’ by testing interaction between feature vs. treatment (usually cutoff of interaction term $p\text{-value} < 0.05$), they may still have subtle different effect size in their prognostic value for the placebo arm vs. the Sunitinib arm. Even though this subtle difference for the two arms is not statistically significant, in aggregate, these features may accumulate meaningful predictive information in the multivariate model, which is also more likely to be transferable to other independent cohorts than individual predictive features.

Finally, the STRAC14 signature, which is actually the true prognostic signature since it is focused on the placebo arm only and is presented in Figure S2, does not overlap at all with the STRACT11

Minor comments:

1. In the abstract: colon should be replaced with a comma and an “and”.

Response: No colon is now used in the abstract as suggested.

2. I would be careful about conclusions about “trends” if CI crosses 1.

Response: we appreciate the Reviewer’s comment and have revised some of the conclusion statement, in particular regarding the point #5 raised above, and have replaced the 5 instances where the term “trend“ was utilized to be more meaningful.

3. I would suggest matching the exact signature names from Fig 3b to the KMs on the same Figure (including their prefixes).

Response: We thank the reviewer for the suggestion and the signature names have been made consistent across all relevant figures.

Reviewer #3 (Remarks to the Author): Expert in renal cell carcinoma clinical research and sunitinib therapy

In this paper, Dr Motzer and coll. report the results of an integrated multi-omics tumor analysis

of 171 patients treated with sunitinib or placebo within the S-TRAC trial, which tested the adjuvant role of sunitinib in high-risk patients following nephrectomy, in order to identify new molecular subtypes associated to these patients and potential new targets.

The study identified new gene expression signatures (GES), including the one consisting of 11 genes defined STRAC11 (derived from the sunitinib-treated population). It also investigated and tested the predictive/prognostic role of gene panels established in the advanced metastatic setting studies IMmotion 150 and Javelin RCC 101, in which sunitinib was used as a comparator treatment, since at the time of the study it was a standard of care for advanced/metastatic stage disease.

The results confirm the landscape of mutations already reported by the TCGA, with VHL, PBRM1, SETD2 and BAP1 as the most frequently mutated in localized RCC. Interestingly, this population resulted enriched in mTOR mutations, implicated in the control of angiogenesis, and that correlated with poor prognosis.

New mutations in THEMIS, WDFY4 or CSPG4 (genes related to T-cell activation, maturation and proliferation), were found to correlate with longer DFS in sunitinib-treated patients. In this regard, it would be interesting to know whether there are corresponding differences in the expression of these genes in patients receiving placebo compared to sunitinib, if feasible, to better distinguish between their (eventual) prognostic and/or predictive impact.

Response: We thank the reviewer for this comment and suggestion which was also brought up by Reviewer #2. The distribution of tumor purity was not significantly different between the placebo and sunitinib arms (t -test; $p=0.6$) and the gene expression

levels were similar between the 2 arms. We have addressed this in response to item #2 above and have added some additional information in the Methods section.

The authors also found that high tumor mutational burden (TMB) correlates with a better outcome only in patients receiving the placebo and not in those treated with sunitinib. It could be of interest to explain if this evidence is the effect of sunitinib (which is able to abrogate such predictive effect of TMB) or it is an epiphenomenon related to a confounding variable, such as a greater activation of genes related to T-cells activation and maturation in the sunitinib arm, as they previously reported. Again, the possible prognostic implication of TMB should be extensively discussed by the authors.

Response: High TMB is now established as a poor prognosis factor in mRCC; here, we observed that sunitinib treatment reduces/limits this effect and as alluded to by the reviewer, this is potentially correlated with the fact that the patients with high tumor CD8 positivity derived greater benefits from treatment as previously reported in this same cohort of patient in reference #11. The authors have added some discussion elements at the end of the paragraph related to the DNA sequencing.

Authors report that two signatures (IMmotion 150 and Javelin Renal 101) are both related to different prognosis when used in the S-TRAC population, however, they do not explain why the significance was lost when the analysis was conducted only in placebo or sunitinib treated patients (as shown in figure 3b). A comment should be added.

Response: Regarding the two Angio signatures, both are prognostic, whereby patients with high expression of the signatures have longer DFS. A similar observation could be made for the IMmotion 150 Teff signature and the Javelin Renal 101 immune signature; however, the confidence intervals were wide and crossed the 1 and thus none reached statistical significance. These points have been clarified in the Results section and some perspective was added to the Discussion.

Along with STRAC-11, authors report two other gene signatures defined STRAC13 and STRAC14, which were also “validated” in the TCGA-KIRC and in the JAVELIN Renal 101. Authors found that STRAC-11 did not differentiate DFS in the axitinib-avelumab treated patients within the Javelin Renal 101 population; the same conclusions are true also for STRAC14 and STRAC13. This paragraph is a bit confusing and it is difficult to understand the ranking among the GES and the reasons to prefer STRAC11 compared to STRAC13 or STRAC14. Authors are invited to clarify this part, reporting pros and cons of each panel and moving the less relevant results to Supplementary material.

Response: The authors thank the reviewer for this comment. The STRAC11 GES was established to identify a group of patients who strong benefited from the sunitinib treatment beyond the UISS clinicopathologic classification while the STARC 14 is based on the placebo arm and thus is actually a prognosis signature. This has been further clarified in the revised manuscript. The figures associated with the STRAC14 and STRAC13 results are currently included in the supplemental material.

The authors have also found and proposed MERTK and TDO2 as two potential new therapeutic

targets in RCC. This is very preliminary and speculative conclusion at this stage and it would be important to analyse and describe also their prognostic role.

In the Discussion, authors should better explain the limits of the panels used (IMmotion 150 and Javelin Renal 101), if they have been previously studied together and what are the results. It is also important to specify that cut-offs are always different because they are based on median values of the studied populations.

Finally, this paper should be put in the context of immunotherapy, considering the classification used for patients eligible for adjuvant therapy in the Keynote-564 trial. It could be of interest to evaluate whether the incidence of mutations is different, according to risk-class populations.

Response: The authors thank the reviewer for these comments. Regarding the previously established and published signatures, the discovery and verification were carried out in the context of advanced/metastatic RCC. We agree that cut-offs are actually not set and will continue to vary until a prospective validation is performed, which is unlikely to occur. For the immunotherapy contextualization, as suggested, reference to the Keynote-564 study outcome and recent approval of pembrolizumab have been added in the discussion.

It is not appropriate the definition “a trend”, when reporting results near the value of $p < 0.05$. Although interesting and stimulating, the findings of this study are mainly descriptive of the landscapes of genes altered in the study population, and do not provide clues for a direct translation into practice. The same authors honestly underline other limits of the study, including: 1) the relative role of angiogenic signature and phenotype, demonstrated by the fact that adjuvant therapy with sunitinib does not prevent metastasis, and the greater relevance of

tumor environment and immune response. 2) the fact that the great majority of patients who recurred had distant metastases, limiting the possibility to draw conclusions on the local progression; 3) the low percent of patients for whom adequate tumor tissue was available (only 27.8% of the total study population), which greatly limits the power of the analysis.

Response: The authors agree with the reviewer's comments. The 5 instances where the term "trend" was utilized have been reconsidered and modified to be more accurate.

In addition, probably due to the need to comply with the constraints of words for publication in Nat Comm, the text is very often difficult to read with too many data from different studies and data set, finally resulting a bit confounding.

Response: The authors thank the reviewer for this comment and have tried to revise the manuscript to clarify as many items as possible within the constraints of the manuscript.

REVIEWER COMMENTS

Reviewer #1 (Remarks to the Author):

I would like to thank the authors for their responses and I have no more comments concerning this study. It is an excellent piece of work that will be beneficial to the field.

Reviewer #2 (Remarks to the Author):

Authors partially answered my comments, but some major concerns remain.

My biggest not-addressed concern is that the authors cannot demonstrate that their 11-gene signature is a signature specific to the treatment arm and not a signature of general aggressiveness. One solution was proposed previously – but it seems that this signature also worked in not-treatment arm. If authors propose that this signature works in BOTH arms, but is still differential between the arms (for example, significantly higher in one arm compared to the other) this needs to be carefully shown and discussed. I would suggest showing results for the 11-gene signature in both arms separately and carefully discussing the differences (however, authors might come up with a more creative solution). Otherwise, the conclusion that this is treatment-arm only needs to be removed.

Another concern is adjustment for other co-variates. I would suggest performing stratified analysis, where a dataset (this should be done for each arm, depending on a question of interest) is divided into female and male sub-sets and performance is evaluated separately on each – indicating if the model performance depends on gender or not. Same would apply for the age categories (whatever age categories make sense in this disease/literature or age median).

Reviewer #3 (Remarks to the Author):

The authors have answered to and clarified all the points addressed by the reviewers. The new updated version is improved and more informative. I am personally satisfied and approve its publication in the present form.

Reviewer comments and author response

Reviewer #1 (Remarks to the Author):

I would like to thank the authors for their responses and I have no more comments concerning this study. It is an excellent piece of work that will be beneficial to the field.

The authors would like to thank the reviewer for their time and valuable feedback improving this manuscript.

Reviewer #2 (Remarks to the Author):

Authors partially answered my comments, but some major concerns remain.

My biggest not-addressed concern is that the authors cannot demonstrate that their 11-gene signature is a signature is specific to the treatment arm and not a signature of general aggressiveness. One solution was proposed previously – but it seems that this signature also worked in not-treatment arm. If authors propose that this signature works in BOTH arms, but is still differential between the arms (for example, significantly higher in one arm compared to the other) this needs to be carefully shown and discussed. I would suggest showing results for the 11-gene signature in both arms separately and carefully discussing the differences (however, authors might come up with a more creative solution). Otherwise, the conclusion that this is treatment-arm only needs to be removed.

The authors understand the concerns of the reviewer and will therefore include a new figure in the Panel #5, substituting the current stratification by the 11-gene signature in the Overall population with that of the 11-gene signature stratification in the Placebo arm.

The associated text in the Results section will be changed accordingly:

In the placebo arm, low and high STRAC11 GES expression groups had similar DFS (HR 0.8 [95% CI: 0.36–1.82]; **Figure 5B**).

Another concern is adjustment for other co-variates. I would suggest performing stratified analysis, where a dataset (this should be done for each arm, depending on a question of interest) is divided into female and male sub-sets and performance is evaluated separately on each – indicating if the model performance depends on gender or not. Same would apply for the age categories (whatever age categories make sense in this disease/literature or age median).

We thank the reviewer for this suggestion. Accordingly, we performed a gender subset analysis of the STRAC-11 signature. As shown below, in both genders, there were similar and consistent patterns. No difference was observed in this analysis between the 2 genders, and the HRs were above 15 in both genders.

Male:

STRAC11_STUENT_STRAT Disease Specific Survival DFS_SUTENT

Type	Statistical Association
Statistic name	cox.regression
Statistic value	27.2412
p-value	1.80e-7
q-value	1.80e-7
Confidence	0.66
Hazard Ratio	17.9084
Model formula	Disease_Specific_Survival_DFS_SUTENT ~ STRAC11_STUENT_STRAT

Female:

STRAC11_STUENT_STRAT Disease Specific Survival DFS_SUTENT

Type	Statistical Association
Statistic name	cox.regression
Statistic value	14.3909
p-value	0.0001
q-value	0.0001
Confidence	0.5242
Hazard Ratio	25.6701
Model formula	Disease_Specific_Survival_DFS_SUTENT ~ STRAC11_STUENT_STRAT

With regard to the age category, we stratified based on the <65 and ≥ 65 years categories and analyzed the performance of the STRAC-11 signature in both. As shown below, in both age groups, there were similar and consistent patterns, and the HRs were greater than 15.

<65 years:

STRAC11_STUENT_STRAT Disease Specific Survival DFS_SUTENT

Type	Statistical Association
Statistic name	cox.regression
Statistic value	33.5819
p-value	6.83e-9
q-value	6.83e-9
Confidence	0.7015
Hazard Ratio	17.3314
Model formula	Disease_Specific_Survival_DFS_SUTENT ~ STRAC11_STUENT_STRAT

≥65 years:

STRAC11_STUENT_STRAT Disease Specific Survival DFS_SUTENT

Type	Statistical Association
Statistic name	cox.regression
Statistic value	8.2299
p-value	0.0041
q-value	0.0041
Confidence	0.407
Hazard Ratio	2319346540.1909
Model formula	Disease_Specific_Survival_DFS_SUTENT ~ STRAC11_STUENT_STRAT

As these and previous covariates were not influencing the performance of the signature, the manuscript has not been modified to include these results.

Reviewer #3 (Remarks to the Author):

The authors have answered to and clarified all the points addressed by the reviewers. The new updated version is improved and more informative. I am personally satisfied and approve its publication in the present form.

The authors would like to thank the reviewer for their time and valuable feedback improving this manuscript.

REVIEWERS' COMMENTS

Reviewer #2 (Remarks to the Author):

Authors have addressed all my concerns.